# Integration-based Extraction and Visualization of Jet Stream Cores

Lukas Bösiger[1], Michael Sprenger[2], Maxi Boettcher[2], Hanna Joos[2], and Tobias Günther[3]

[1]Department of Computer Science, ETH Zurich, Switzerland
[2]Institute for Atmospheric and Climate Science, ETH Zurich, Switzerland
[3]Department of Computer Science, FAU Erlangen-Nürnberg, Germany

**Correspondence:** Tobias Günther (tobias.guenther@fau.de)

**Abstract.** Jet streams are fast three-dimensional coherent air flows that interact with other atmospheric structures such as warm conveyor belts (WCBs) and the tropopause. Individually, these structures have a significant impact on the mid-latitude weather evolution, and the impact of their interaction is still subject of research in the atmospheric sciences. A first step towards a deeper understanding of the meteorological processes is to extract the geometry of jet streams, for which we develop an integration-based feature extraction algorithm. Thus, rather than characterizing jet coreline purely as extremal line structure of wind magnitude, our coreline definition includes a regularization to favor jet corelines that align with the wind vector field. Based on the line geometry, proximity-based filtering can automatically detect potential interactions between WCBs and jets, and results of an automatic detection of split and merge events of jets can be visualized in relation to the tropopause. Taking ERA5 reanalysis data as input, we first extract jet stream corelines using an integration-based predictor-corrector approach that admits momentarily weak air streams. Using WCB trajectories and the tropopause geometry as context, we visualize individual cases, showing how WCBs influence the acceleration and displacement of jet streams, and how the tropopause behaves near split and merge locations of jets. Multiple geographical projections, slicing, as well as direct and indirect volume rendering further support the interactive analysis. Using our tool, we obtained a new perspective onto the three-dimensional jet movement, which can stimulate follow-up research.

## 1 Introduction

Meteorologists face the challenge of making sense out of highly complex atmospheric data, which are large, three-dimensional, time-dependent, and contain multiple fields. These data are traditionally examined in 2D (Rautenhaus et al., 2018), with more modern approaches transitioning towards 3D (Li et al., 2019; Rautenhaus et al., 2015b). This is because the atmospheric circulation contains and is governed by numerous three-dimensional flow features and processes that significantly impact the weather (Nielsen-Gammon, 2001; Harnik et al., 2016). For example near the tropopause, the boundary between the troposphere and the stratosphere, we can find fast coherent air streams called jet streams (Koch et al., 2006; Schiemann et al., 2009). The wind speed of jet streams often exceeds $50\,\mathrm{m\,s^{-1}}$ at their core, as they meander from west to east around the planet (Ahrens and Henson, 2018). Jet streams can interact with the tropopause as well as with outflows of warm conveyor belts (WCBs), which are strongly slantwise and coherently ascending air streams within extratropical cyclones that diverge at the tropopause (Madonna et al., 2014). Both jets and WCBs are well known to play a major role in mid-latitude weather evolution and predictability (Harnik

et al., 2016; Manney et al., 2014; Manney and Hegglin, 2018). However, how much they influence each other and which physical processes are involved in this interaction is not yet fully explored in the atmospheric sciences.

In this paper, we aim for an explicit extraction of jet stream coreline geometry, which enables automatic detection of proximity to WCBs and allows for the automatic extraction of split and merge events. We visually investigate the shape of the three-dimensional tropopause near the split and merge events by visualizing the structures in 3D. This requires a robust extraction algorithm emitting long jet stream corelines that are temporally stable and long enough to find split and merge locations. To this end, we extend the recent work of Kern et al. (2017), who developed a local jet core extraction algorithm, which may result in fragmented coreline segments when the velocity magnitude temporally falls below a required threshold. Thus, instead, we extract the jet stream as an instantaneous flow feature from the atmospheric data using a predictor-corrector approach (Banks and Singer, 1994) that follows the flow and corrects the trajectories towards the ridge lines of the wind magnitude field. This way, we can balance between two constraints: maximizing the wind speed and aligning with the flow direction. The latter serves as regularization to prevent unnaturally bent ridge lines caused by a low vertical resolution. Thereby, jet stream corelines are allowed to fall below the magnitude threshold for a user-defined period of time. To display context and to locate possible interactions with other atmospheric structures, we extract the tropopause based on its dynamical definition, and selected WCBs as pathlines fulfilling a necessary ascent criterion in the neighbourhood of extratropical cyclones. In an interactive visualization system, the proximity and interaction of these features can be explored using temporal animations, slicing and the inspection of related scalar variables. Since WCBs are Lagrangian structures arising from integration over a time window, we intersect their paths in space-time with the time slice selected for the view of the instantaneous tropopause and jet streams, resulting in a set of WCB positions at the time instance of the jet. By spatially filtering these positions by their proximity to jets, we identify locations at which the structures are close to each other. Using our interactive tool, multiple cases are studied and interpreted, including the acceleration of jets in the vicinity of WCBs, the movement of jets, the path of jets along tropopause folds, as well as splitting and merging events of multiple jets at the tropopause. We evaluate the parameter sensitivity and the performance of the jet stream extraction. The feature extraction pipeline opens new opportunities for further research in atmospheric sciences, and the visual analysis tool sheds light on three-dimensional processes that are difficult to grasp in 2D slices. In summary, we contribute:

- an integration-based jet core extraction algorithm that takes a predictor-corrector approach (Banks and Singer, 1994) to extract longer jet stream corelines as approximate ridge lines in the wind magnitude field, while allowing the feature lines to be regularized to align with the wind vector field,

- an automatic extraction of locations at which jet streams split or merge, which are classified based on the integration direction of the jets,

- an interactive visualization of the extracted line geometry, which enables us to study situations where WCBs approach a jet, as well as jet splits and merges at the folds of the tropopause.

The paper is structured as follows. Section 2 places the work among related work and explains relevant meteorological concepts that later inform the feature extraction. Section 3 introduces our novel predictor-corrector based jet stream extraction algorithm,

compares the approach with a local jet core extraction, and studies the performance and parameters of the proposed method. Section 4 integrates the jet core extraction into an interactive visualization system that allows the user to study the co-occurence of jets with other meteorological features. In Section 5, the jet stream cores are visualized along warm conveyor belts and the tropopause to study the behavior of jets in the vicinity of other meteorological structures, e.g., tropopause folds. Section 6 concludes the work and outlines opportunities for further research.

## 2  Meteorological Concepts

We begin with a brief introduction of the meteorological background and a summary of recent work on the visualization of meteorological features. For a comprehensive introduction to meteorological visualization, we refer to Rautenhaus et al. (2018).

### 2.1  Potential Vorticity

Potential vorticity (PV) is a scalar quantity that measures the rotation of the air enclosed between two isolevels of potential
temperature. It is a common diagnostic tool, as it is preserved during advection in adiabatic and frictionless flow conditions. Following Hoskins et al. (1985), we use potential vorticity PV, here defined for all quantities in isobaric coordinates, i.e., in pressure levels:

$$\text{PV} = -g \cdot (f \cdot \mathbf{k} + \nabla \times \mathbf{v}) \cdot \nabla \theta \tag{1}$$

where $g$ denotes the acceleration of gravity, $f = 2\Omega \sin(\phi)$ is the Coriolis parameter, with $\Omega = 7.29 \cdot 10^{-5} \text{ rad} \cdot \text{s}^{-1}$ being the
angular velocity of the Earth, $\phi$ denoting latitudes, and $\theta$ being the potential temperature. PV is measured in potential vorticity units (pvu) where $1 \text{ pvu} = 10^{-6} \text{ K m}^2 \text{ kg}^{-1} \text{ s}^{-1}$. Recently, Bader et al. (2020) extracted and visualized banners of potential vorticity, detaching from orographic mountain peaks.

### 2.2  Tropopause

The tropopause is the atmospheric boundary between the troposphere and the stratosphere (Holton et al., 1995; Stohl et al.,
2003). The tropopause can exist anywhere between 70 hPa (around 18 km) and 400 hPa (around 6 km), c.f. Dameris (2015), and might regionally drop even lower (Lillo et al., 2021). The tropopause altitude is generally highest in the tropics, lowest near the poles, and drops sharply across the subtropical jet. Unlike its common illustration in text books, it is far from being a smooth surface. Distinct structures, such as tropopause folds and stratospheric intrusions, frequently occur and give the tropopause a complex 3D geometry (Danielsen, 1968; Shapiro, 1980; Škerlak et al., 2015). These structures can have an impact on a wide
range of weather phenomena on the Earth surface (Nielsen-Gammon, 2001). For instance, the evolution of surface cyclones is often associated, or even triggered, by localized perturbations of the tropopause. Based on the potential vorticity, we seize the dynamic definition of the tropopause, as it captures complex perturbations of the troposphere-stratosphere interface. To define the tropopause, different thresholds of PV iso-surfaces are used, typically between 1 to 5 pvu (Highwood et al., 2000; Schoeberl, 2004; Kunz et al., 2011; Dameris, 2015), depending on the region and the analysis task. We refer to Nielsen-Gammon (2001) for

visualizations of 20 years of tropopause data, defined in terms of potential vorticity, and to Škerlak et al. (2015) for the subtleties
in extracting the dynamical tropopause from numerical weather prediction (NWP) data.

## 2.3  Jet Streams

Jet streams are fast and coherent air streams in the upper troposphere and lower stratosphere, which are hundreds of kilometers
wide and only a few kilometers thick. The center of the jet stream, which is called *jet core*, often exceeds $50\,\mathrm{m\,s^{-1}}$ and can

occasionally reach wind magnitudes of $100\,\mathrm{m\,s^{-1}}$ (Koch et al., 2006; Schiemann et al., 2009; Ahrens and Henson, 2018). Jet
streams and their variation are of high interest, cf. (Manney et al., 2014; Manney and Hegglin, 2018), for example in relation
to extreme weather events (Harnik et al., 2016). By the definition of the World Meteorological Organization (WMO) the jet
stream is defined as "*Flat tubular, quasi-horizontal, current of air generally near the tropopause, whose axis is along a line of
maximum speed and which is characterized by great speeds and strong vertical and horizontal wind shears.*" (WMO, 1992).

The heart of the jet stream, the jet core, is defined as "*Line along which the wind speeds are maximal both in the vertical and in
the horizontal.*" (WMO, 1992).

Several approaches have been developed to automatically extract jet streams, cf. Maher et al. (2019) for a comprehensive
overview. A common approach is to threshold the wind magnitude in a certain height range, as for example done by Limbach
et al. (2011) and Martius (2014). An additional constraint is the assumption that the flow is oriented eastwards, cf. Schiemann

et al. (2009). Alternatively, Archer and Caldeira (2008) locally considered mass and mass-flux weighted averages to detect
the jets. To compute the average jet wind speed, Koch et al. (2006) counted grid points that exceed a wind speed of $30\,\mathrm{m\,s^{-1}}$
between 100 and 400 hPa. Once grid points are identified that belong to jet cores, they can be further classified into different
types of jets. The most important ones being the polar and subtropical jet. This classification, however, is not trivial, since there
is a continuous spectrum of jet characteristics (Lee and kyung Kim, 2003; Manney et al., 2014, 2021; Winters et al., 2020).

Pena-Ortiz et al. (2013) noted that attempting to distinguish polar and subtropical jets by latitude was commonly unsuccessful;
Manney et al. (2011, 2014) found that using a simple latitude criterion was only useful for very broad climatological studies,
and Manney and Hegglin (2018) introduced a more physically-based method of distinguishing subtropical and polar jets based
on tropopause height changes across the jet region. Winters et al. (2020) (and references therein) distinguish subtropical and
polar jets by identifying them in different isentropic layers, and show clear instances of them merging into jets with hybrid

characteristics. Manney et al. (2011) identified sectional extrema in wind speed above $40\,\mathrm{m\,s^{-1}}$ inside latitude-altitude slices. If
more than two local extrema appeared within the $30\,\mathrm{m\,s^{-1}}$ isocontour at a distance of more than 15° latitude apart or when
the wind speed between the two local extrema drops below $25\,\mathrm{m\,s^{-1}}$, the extrema were considered to be separate jet stream
cores. This approach was applied by Manney et al. (2014) in a higher-resolution reanalysis. The approach was later extended
by Manney and Hegglin (2018); Manney et al. (2021) to include a physically-based distinction between subtropical and polar

jets. Spensberger et al. (2017) identified upper-tropospheric jet corelines as locations with vanishing wind shear orthogonal to
the wind direction and applied the approach to study jet variability on the Northern and Southern Hemispere (Spensberger and
Spengler, 2020). Maher et al. (2019) introduced the tropopause gradient method, which extracts jet cores by detecting turning

points in the potential temperature observed along the 2 pvu isocontour. Winters et al. (2020) investigated polar-subtropical jet superpositions. Kern and Westermann (2019) studied clustering methods for ensembles of jet corelines.

Most of the methods above assume certain characteristics of the jet. For example that the jet moves from west to east, or that it is continuous (Barton and Ellis, 2009). Kern et al. (2017) developed an approach which does not assume such characteristics and detects the jet at all levels and directions equally. By defining a local coordinate system that is composed of the horizontal wind direction $\mathbf{s} = (u, v, 0)$ (with $u$ being oriented eastward and $v$ being oriented northward), the normal direction $\mathbf{n} = (-v, u, 0)$ and the vertical axis $\mathbf{k} = (0, 0, 1)$, jet core lines appear as sectional extrema of the horizontal velocity magnitude $s(\mathbf{x}) = \|\mathbf{s}\|$ in the

$\mathbf{n}$-$\mathbf{k}$ plane by intersecting the two isosurfaces:

$$\frac{\partial s}{\partial \mathbf{n}} = 0 \qquad \frac{\partial s}{\partial \mathbf{z}} = 0 \tag{2}$$

which are further filtered by an eigenanalysis of the Hessian of $s(\mathbf{x})$. While Kern et al. (2017) used a specialized variant of the marching cubes algorithm (Ljung and Ynnerman, 2003) to find solutions to the implicit equations in Eq. (2), we rephrase this problem for later comparison into a standard parallel vectors (Peikert and Roth, 1999) problem for which several local (Peikert

and Roth, 1999), integration-based (Van Gelder and Pang, 2009; Weinkauf et al., 2011) and implicit solvers (Witschi and Günther, 2020) exist, cf. Günther and Theisel (2018):

$$\begin{pmatrix} \partial s/\partial \mathbf{n} \\ \partial s/\partial \mathbf{z} \\ 0 \end{pmatrix} \parallel \begin{pmatrix} 0 \\ 0 \\ 1 \end{pmatrix} \quad \Leftrightarrow \quad \begin{pmatrix} \partial s/\partial \mathbf{n} \\ \partial s/\partial \mathbf{z} \\ 0 \end{pmatrix} \times \begin{pmatrix} 0 \\ 0 \\ 1 \end{pmatrix} = \begin{pmatrix} 0 \\ 0 \\ 0 \end{pmatrix} \quad \Leftrightarrow \quad \begin{pmatrix} \partial s/\partial \mathbf{z} \\ \partial s/\partial \mathbf{n} \\ 0 \end{pmatrix} = \begin{pmatrix} 0 \\ 0 \\ 0 \end{pmatrix} \tag{3}$$

The symbol $\parallel$ denotes the parallel vectors operator (Peikert and Roth, 1999), which receives two vector fields as input and produces the set of points at which the two given vector fields are parallel. The two vectors are parallel if their cross product

vanishes to zero. Applying the cross product results in three equations: the two equations from Eq. (2) and $0 = 0$. Since this is a local feature extraction method, the resulting jet cores are often short and have to be heuristically connected. Especially in combination with a wind magnitude filter, the lines can decay into pieces if no global ideal threshold exists. This problem can be reduced by prior smoothing of the fields, which, however, affects the precise location of the feature. Rather than extracting lines locally, we choose an integration-based approach to extract the jet core lines using a predictor-corrector algorithm (Banks and

Singer, 1994). Our aim is to not only extract core line segments, but also to keep track of the connections between them. We do this by allowing the jet core to be weaker than the strength threshold for a limited number of integration steps. In addition, we identify splitting and merging events of jet corelines.

## 2.4   Warm Conveyor Belts (WCBs)

Cyclones are important components of the climate system. They are responsible for a major fraction of the meridional moisture

and heat transport, and produce most of the precipitation in the mid-latitudes (Wernli and Schwierz, 2006; Schultz et al., 2019). Further, they clean the atmospheric boundary layer from aerosols and pollution (Eckhardt et al., 2004). The warm conveyor belt (WCB) is one out of three important air streams found in extra-tropical cyclones. The others are the dry intrusion (Raveh-Rubin,

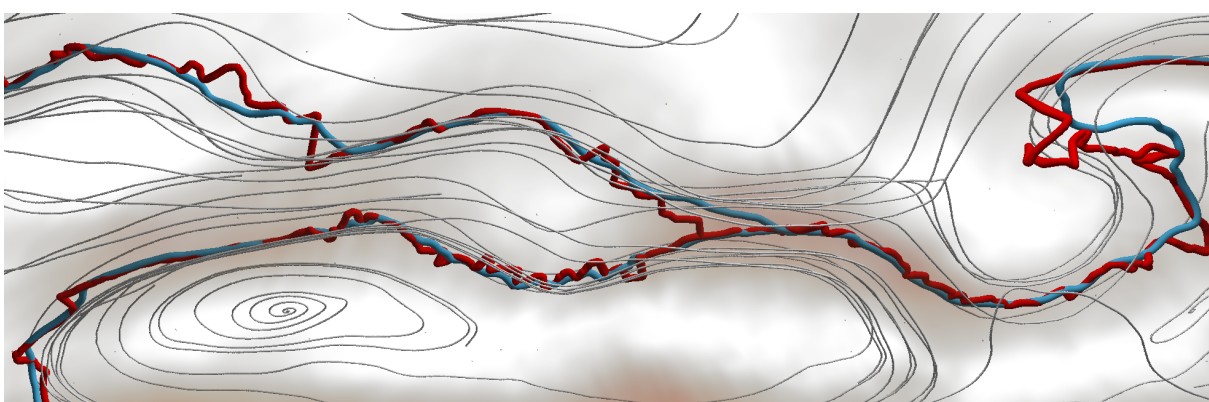

**Figure 1.** Extremal lines (red lines) of the wind magnitude are sensitive to the data discretization and thus do not follow the streamlines (gray lines) of the wind field. In contrast, our jet corelines (blue lines) balance between extremal lines and flow alignment. All lines are traced in 3D and may therefore intersect on the viewport. Locations with wind speed between 10 and 40 ms$^{-1}$ are highlighted with gray-red color in the background, showing that jet corelines reside in high velocity regions. 01.09.2016 00:00.

2017) and the cold conveyor belt (Madonna et al., 2014). The WCB is responsible for strong cloud formation, for precipitation as well as for modifying the potential vorticity (Madonna et al., 2014). Further, WCBs influence the evolution of cyclones. For example, a strong WCB next to an extra-tropical cyclone can hint at rapid storm development (Binder et al., 2016). Also they can be used to characterize the geographical distribution of cyclones (Eckhardt et al., 2004). They are also highly relevant to numerical weather prediction, because they can be a major source and magnifier of forecast uncertainty (Rodwell et al., 2018). WCBs usually originate in the moist subtropical marine boundary layer between 20 and 45° latitude (Wernli, 1997). In the Northern Hemisphere, most of the WCBs appear close to a center of a cyclone and they are more common in winter than in summer. In the Southern Hemisphere, the WCBs vary less and there are no WCBs over polar regions (Madonna et al., 2014). To find WCBs, a Lagrangian selection criterion can be used based on the fact that WCBs ascend strongly, typically by more than 600 hPa with 48 hours. Similar criteria have been used by Wernli (1997), Eckhardt et al. (2004) and Madonna et al. (2014). For example, Eckhardt et al. (2004) extracted 15 years of WCB trajectories starting at 500 m above ground level. They extracted only the trajectories which in two days traveled northward and ascended at least 60% of the zonally and climatologically average tropopause height. Rautenhaus et al. (2015a) estimated the probability for the existence of WCBs from particle trajectories in an ensemble of flows and visualized WCBs in 3D. In addition to the slantwise and slow ascent, recent studies considered the rapid, convective transport that can be embedded in the WCB (Oertel et al., 2019).

## 3 Integration-based Jet Stream Extraction

### 3.1 Data

For our analysis, we used the ERA5 reanalysis data (Hersbach et al., 2020), provided by the European Centre for Medium-Range Weather Forecasts (ECMWF). We use hourly data from 01.09.2016 00:00 until 31.10.2016 23:00, which coincides with the field

campaign NAWDEX (Schäfler et al., 2018). The spatial data is stored on a regular grid with dimensions $720 \times 361 \times 98$ for three-dimensional fields and $720 \times 361$ for two-dimensional fields. The vertical dimension is given on hybrid sigma pressure levels (Eckermann, 2009), spanning a range from 27.8 hPa to 1039.9 hPa. For jet core extraction in our two month data set, we consider the range from 190 hPa to 350 hPa, in which the jets can be expected. Jets may very well occur outside of this range for other time periods, as demonstrated by Manney et al. (2014) both below and above our chosen thresholds. Thus, depending on the spatial and temporal region of interest, the thresholds should be adapted. The horizontal grid spacing is $0.5°$ in lat/lon and for computational convenience, we resampled the height levels onto uniformly-spaced 10 hPa pressure levels. For our data, the regridding produces approximately 10 times more vertical samples, which consumes additional memory. This could be avoided by working directly on the hybrid model level data, which requires adjustments in the calculation of partial derivatives and in the interpolation. In principle, it would also be possible to perform the jet stream coreline extraction in other coordinate systems, for example in isentropic coordinates where the vertical levels have equal potential temperature.

We refer the interested reader to the works of Kern et al. (2017) and Kern and Westermann (2019), who studied jets in the same time period, including an ensemble analysis which is another practical application of explicitly extracted jet stream corelines.

## 3.2 Feature Extraction

Kern et al. (2017) defined jet corelines as ridge lines of the wind magnitude field. For numerical weather data, local Hessian-based ridge detectors (Eberly, 1996) pose numerical challenges in the second-order derivative estimation and the subsequent eigenanalysis, as pointed out by Kern et al. (2017), who locally extracted the ridge line with a first-order method as described earlier in Eq. (2). Especially in a low vertical resolution in the upper atmospheric layers, a pure extremal line definition can suffer from numerical problems, as shown in Fig. 1 (red lines). Since local methods further require heuristics to connect line segments, we apply the predictor-corrector algorithm by Banks and Singer (1994), where we change the underlying scalar and vector field. With this, ridge lines are calculated incrementally by alternating between predictions and corrections. The predictor-corrector approach serves two purposes. First, it allows tracing out those lines leading to longer connected jets. Second, by balancing between predictor and corrector steps, the trajectory can be regularized to follow wind vectors directly during feature extraction, rather than in a post-process. With this, unnaturally bending jet corelines due to discretization problems can be reduced, see Fig. 1 (blue lines).

In the following, we explain the algorithm in detail. For brevity, we omit the time parameter $t$ in the velocity magnitude field $s(\mathbf{x})$ and the *normalized* wind velocity field $\bar{\mathbf{v}}(\mathbf{x}) = \mathbf{v}(\mathbf{x})/||\mathbf{v}(\mathbf{x})||$. We utilize a normalized field, since corelines are extracted per time step. In such an instantaneous flow, normalization results in the same streamline geometry but with more explicit control over the integration speed.

**Initial seed points**

As initial seed points in the first frame, we select all local maximal extremal points $\mathbf{x}_0$ of the wind magnitude field $s(\mathbf{x})$ in Fig. 2a with a wind magnitude of at least $s(\mathbf{x}_0) > 40\,\text{ms}^{-1}$ between 350 hPa (around 8 km from the Earth surface) and 190 hPa

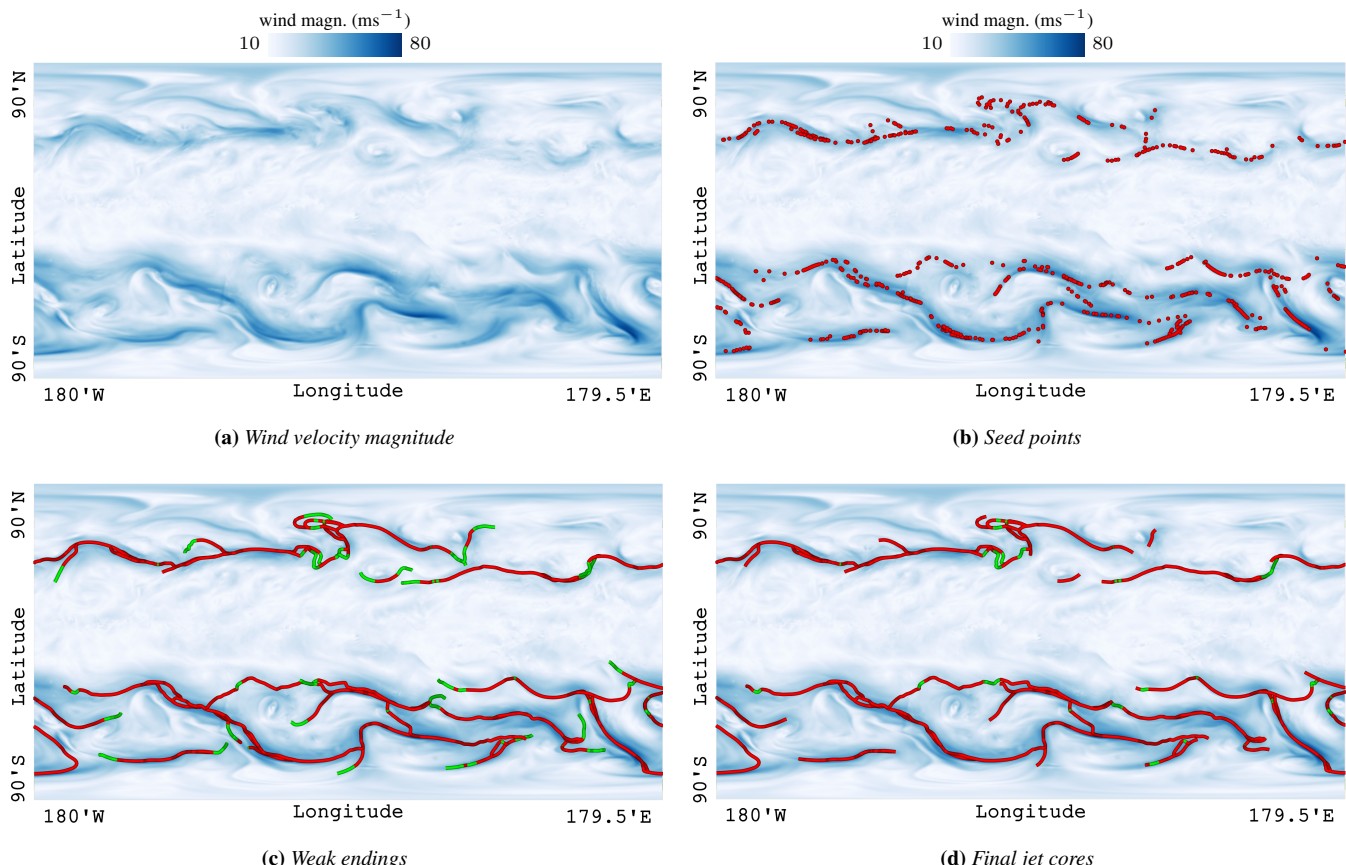

**(a)** *Wind velocity magnitude*

**(b)** *Seed points*

**(c)** *Weak endings*

**(d)** *Final jet cores*

**Figure 2.** (a) shows a slice of the wind magnitude field. White indicates low wind speed, blue high wind speed. (b) same as a) overlaying seed points in red. (c) shows predictor-corrector results with weak endings in green. (d) shows the final result, after the weak endings have been removed. Finally, green segments denote regions at which the velocity magnitude threshold was temporarily not reached. 08.09.2016 12:00.

(around 12 km from the Earth surface). We chose this region and the wind speed threshold empirically for our data set. Note that we use discrete extremal points, i.e., grid points for which all neighboring grid points have a smaller value, in order to avoid the numerical estimation of derivatives. The seed points are shown in Fig. 2b. All seed points are sorted in descending order by the wind magnitude, i.e., we begin our further processing with the global maximum. From each seed point, the jet coreline is traced forward and backward, as explained in the following.

**Prediction**

Assuming that the flow direction is a fair estimate for the jet coreline tangent, the prediction estimates the next point $\widehat{\mathbf{x}}_{i+1}$ of the jet coreline in the direction of the velocity field $\bar{\mathbf{v}}(\mathbf{x})$:

$$\widehat{\mathbf{x}}_{i+1} = \mathbf{x}_i + h \cdot \bar{\mathbf{v}}(\mathbf{x}_i) \tag{4}$$

with $h$ being the integration step size for which we use the grid spacing. In practice, we take a fourth-order Runge-Kutta step instead of an Euler step. The velocity $\bar{\mathbf{v}}(\mathbf{x}_i)$ at point $\mathbf{x}_i$ is trilinearly interpolated from the discrete grid storing the velocity values. The flow direction is not only an estimate for the next vertex location, it also serves as regularization to align lines with the wind vector field.

**Correction**

Since $\widehat{\mathbf{x}}_{i+1}$ might not be precisely on the ridge line of the wind magnitude field, a correction is applied, which moves the point in the plane orthogonal to the flow direction back towards the ridge line:

$$\mathbf{x}_{i+1} = \widehat{\mathbf{x}}_{i+1} + h \cdot \left( \nabla s(\widehat{\mathbf{x}}_{i+1}) - \bar{\mathbf{v}}(\widehat{\mathbf{x}}_{i+1}) \cdot \nabla s(\widehat{\mathbf{x}}_{i+1})^\mathrm{T} \bar{\mathbf{v}}(\widehat{\mathbf{x}}_{i+1}) \right) \tag{5}$$

which is iterated until a maximal number of iterations is reached. The number of correction iterations determines how closely the extracted jet stream will follow a ridge line of the wind magnitude field (high number of iterations) or be tangential to the wind field (low number of iterations). We show different choices later in Section 3.4. In this procedure, the velocity field is normalized, such that the travelled distance in both the prediction and the correction is independent of the wind speed, and is instead controlled by the number of prediction steps $n_{predSteps}$ and the number of correction steps $n_{corrSteps}$ in each predictor-corrector iteration.

**Termination**

Iterating the predictor and corrector step incrementally traces out the jet corelines. We terminate the jet coreline tracing under three conditions: (1) the wind magnitude $s(\mathbf{x})$ remained below a user-defined threshold for more than $n$ prediction steps (to admit line segments that fall below the threshold only for a short amount of time), (2) the angle between two consecutive line segments is larger than $60°$, which is a quality criterion adapted from Kern et al. (2017) to prevent unnatural bending, (3) the new predictor step comes too close to an already computed jet coreline. The latter avoids the duplicate tracing of jet corelines, and allows recording split and merge events as explained below. In contrast, Bader et al. (2020) applied an agglomerative hierarchical clustering to remove duplicates, which required a suitable line distance measure and distance threshold.

**Split and Merge Detection**

The previously mentioned third termination criterion is used to locate coordinates at which jet streams split or merge. If the new point is closer than a user-defined threshold to an already existing jet stream, we record the event. The search for the closest point on the previously traced jet corelines is accelerated by using a kd-tree and performing a nearest-neighbor search, for which we used the library nanoflann (Blanco and Rai, 2014). Empirically, we applied a horizontal distance threshold of $1°$ latitude or longitude and a vertical distance below $5\,\mathrm{hPa}$ as threshold. The type of event (split or merge) is identified by the line orientations. If a jet stream integration approaches an existing jet stream while doing a forward integration, then we mark this location as a merge point. If the jet stream approaches an existing jet stream during a backward integration, then a split point is found.

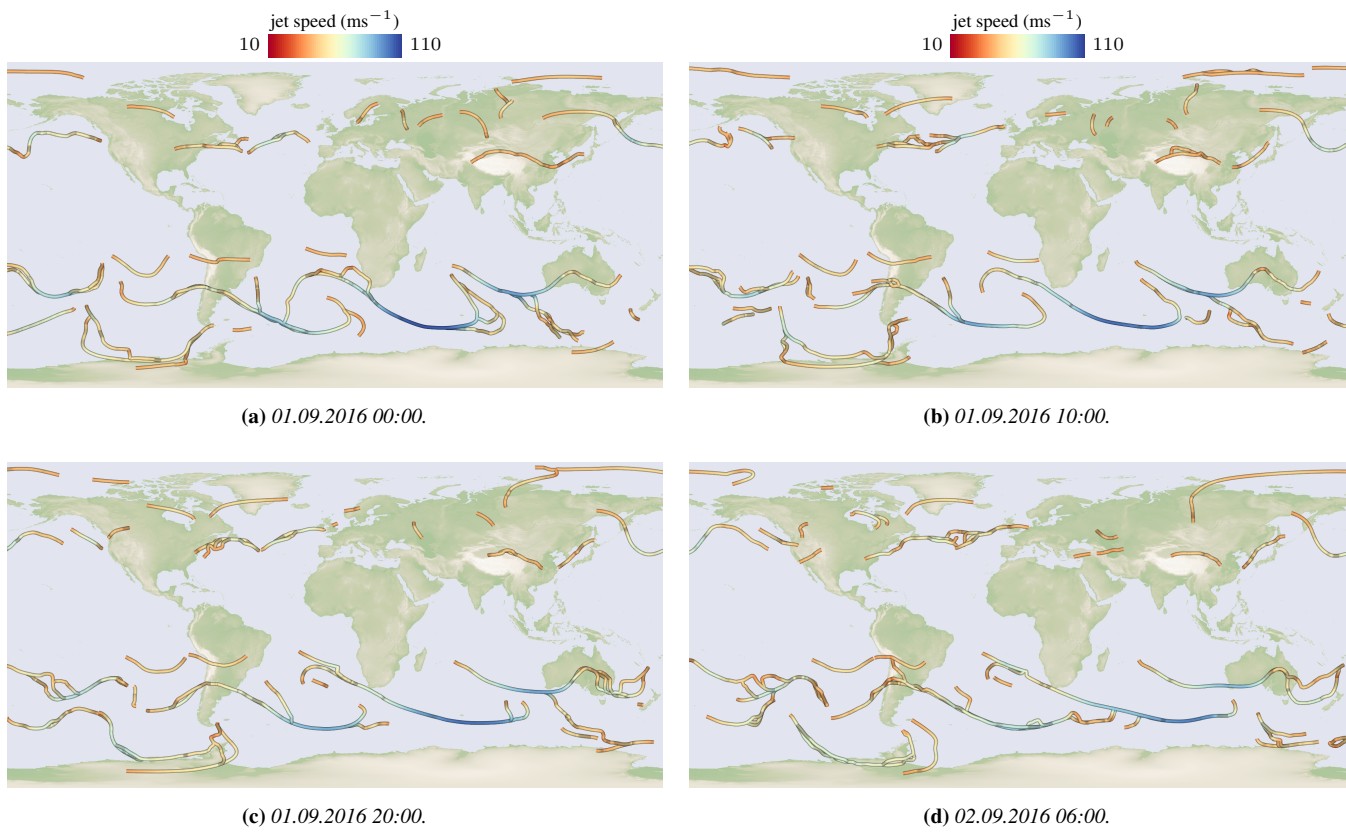

**(a)** *01.09.2016 00:00.*

**(b)** *01.09.2016 10:00.*

**(c)** *01.09.2016 20:00.*

**(d)** *02.09.2016 06:00.*

**Figure 3.** Temporal tracking of the jet stream. Here, shown for 4 time steps with an integration step size of 10 hours between 01.09.2016 00:00 and 02.09.2016 06:00. Especially the fast moving wind speeds (blue structures) are tracked coherently over time.

### Pruning

Since the predictor-corrector algorithm tracks jet corelines even after they fall under the wind magnitude threshold, all lines have an ending which is below the jet stream threshold $\tau_{wind}$. In Fig. 2c, this is indicated with green lines, whereas the red lines actually belong to the jet core. We remove those weak endings in a post-process by pruning end points until the velocity magnitude threshold of the last point is reached, see Fig. 2d.

### Temporal Tracking

In the first frame, we extracted local wind magnitude extrema above the wind magnitude threshold as potential seed points, sorted them by the wind magnitude and incrementally applied the predictor-corrector algorithm for one seed point at a time in descending order. For all subsequent frames, we construct our seed point set differently. For all jet corelines of the previous frame, we first extract the local wind magnitude extrema on the jet cores to form candidates. Each of these points performs a local gradient ascent in the next time step to find a nearby local maximum. This way, the seed point set remains temporally more

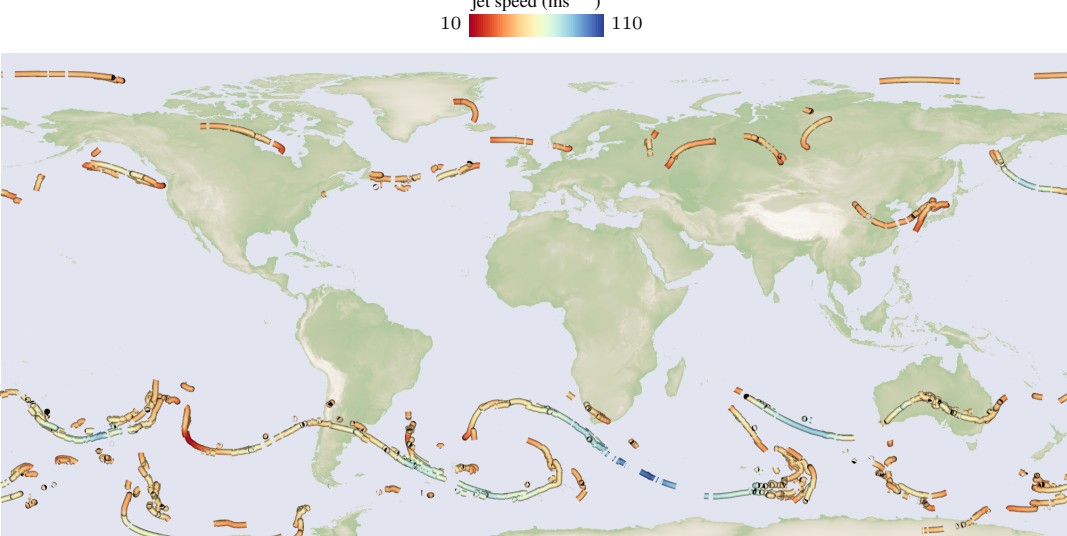

**(a)** *Local extraction using the parallel vectors formulation in Eq.* (3).

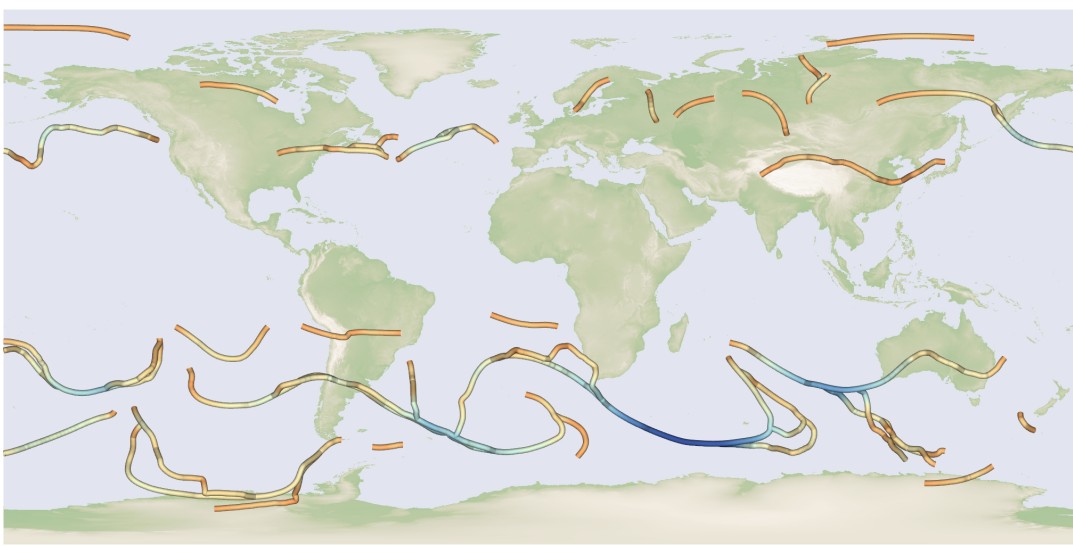

**(b)** *Our predictor-corrector extraction.*

**Figure 4.** Local extraction methods on unsmoothed data (top) may lead to spurious jet corelines that need substantial post-processing and filtering. Below, our method results in continuous curves. Extraction results are shown for 01.09.2016 00:00.

coherent than when it was created anew. Seed points are removed if the new local maximum does not reach the wind magnitude threshold anymore. To not miss jets that are forming in the next time step, we insert local wind magnitude extrema in regions not covered by jets. In Fig. 3, four time steps of an animation are shown, which display the movement of jets and their temporal evolution. We refer to the accompanying video for the full animation.

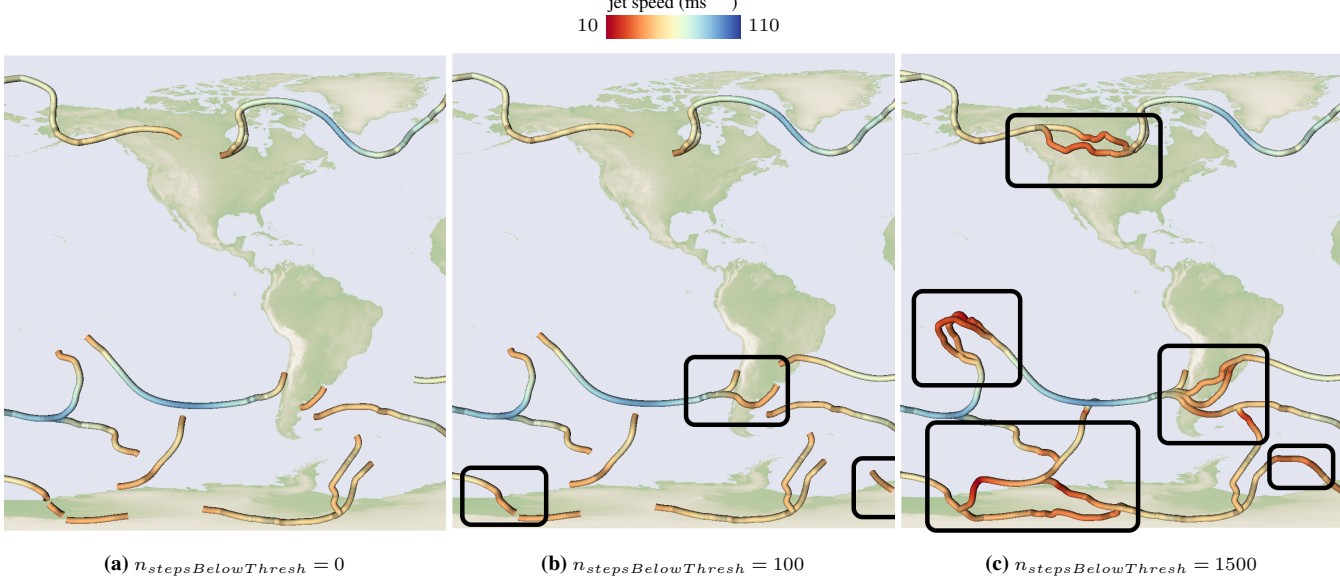

**(a)** $n_{stepsBelowThresh} = 0$      **(b)** $n_{stepsBelowThresh} = 100$      **(c)** $n_{stepsBelowThresh} = 1500$

**Figure 5.** Increasing the number of prediction steps a jet core might remain below the wind magnitude threshold, results in longer connected jet corelines. Here, for 11.09.2016 15:00.

### 3.3 Comparison with Local Method

In numerical simulations, the domain is discretized onto a grid, composed of cells in which attributes are interpolated. Local line extraction methods solve for lines in three steps (Peikert and Roth, 1999). First, intersection points with the cell boundaries are computed numerically per cell. Second, the intersection points are connected to form line segments within the cells, which may fail if intersection points were missed or are duplicated due to numerical reasons. Third, the line segments are connected to continuous lines when the end points of two segments are close enough to each other (within a threshold) and when the

tangent directions at the end points align (up to a certain threshold). The result of this last operation is order-dependent, depends on the numerical accuracy of the first step, and is dependent on thresholds. Fig. 4a gives an example of the parallel vectors formulation in Eq. (3), which produces many small disconnected line pieces, depending on the threshold choices. Further, the lines are restricted to regions in the domain with a velocity magnitude larger than $40\,\mathrm{ms}^{-1}$. In order to produce clean results, local methods often require parameter tuning, filtering (for example by line length), or extensive smoothing of the input fields.

Instead, our integration-based approach in Fig. 4b can naturally grow the lines in direction of the vector field, allowing us to construct long connected jet corelines.

### 3.4 Performance and Parameter Study

In this section, we analyze the performance and discuss parameter choices. The system on which we run the measurements contains an AMD Ryzen 9 3900X CPU. In the preprocess, we resample the data to equidistant pressure levels $10\,\mathrm{hPa}$ apart

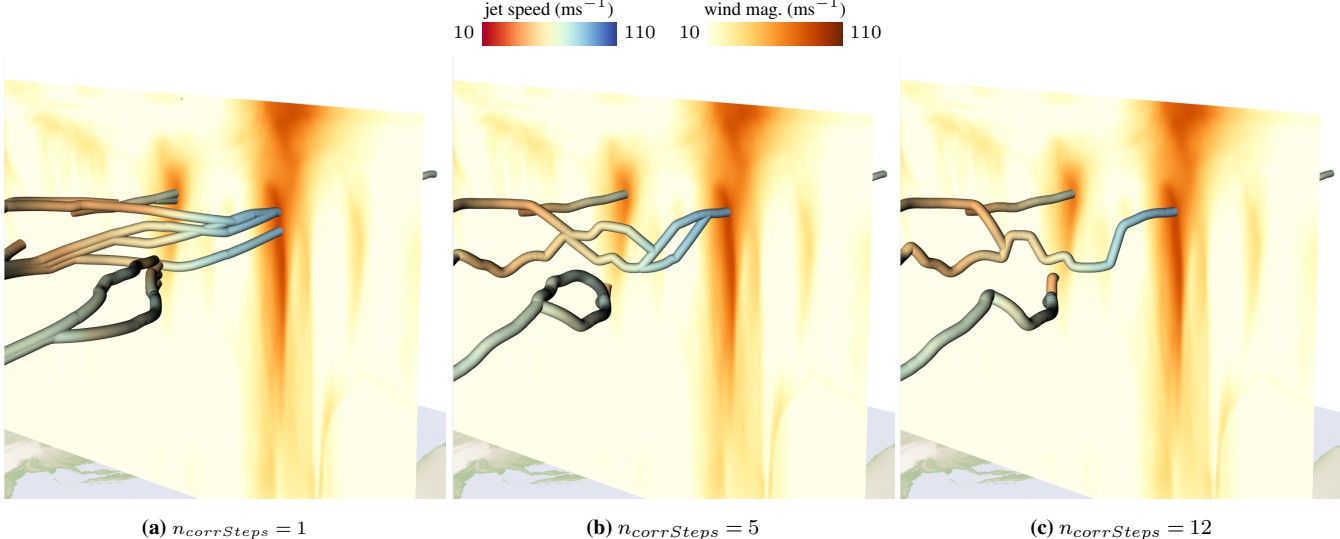

**(a)** $n_{corrSteps} = 1$      **(b)** $n_{corrSteps} = 5$      **(c)** $n_{corrSteps} = 12$

**Figure 6.** Comparison of jet corelines for varying number of correction iterations. The higher the number, the closer the line follows a ridge line as proposed by Kern et al. (2017), which might exhibit higher curvature. Lowering the number of corrector iterations smoothes the line. A cross section of the wind magnitude field shows how well the extracted corelines pass through sectional extrema. Here, for 11.09.2016 15:00.

$(720 \times 361 \times 104)$, which takes on average 28.2 seconds per time step and generates 1.62 GB additional disc space per time step. The WCB preprocessing for all time steps takes 213.6 seconds and generates 0.86 GB additional disc space. On average, the extraction time of jet stream corelines takes about 10 seconds per simulation time step. For a total of two months of hourly simulation data, this leads to a preprocessing time of 12 hours for the whole data set and an additional disc space usage of 2.37 TB, compared to the original data set size of 903 GB.

The default parameters of the jet core extraction and the performance measurements can be seen in Table 1 for time step 01.09.2016 00:00. Increasing $\tau_{wind}$, i.e., the wind magnitude threshold for being classified as jet stream, leads to fewer jet corelines, because fewer vertices meet the filter criterion. In contrast, if chosen smaller, more vertices will pass the threshold filter and will be recognized as belonging to jet corelines. Increasing $n_{stepsBelowThresh}$, i.e., the number of integration steps a core line can be below the velocity magnitude threshold, leads to more connected structures. In Fig. 5, we demonstrate the impact

of varying the number of prediction steps $n_{stepsBelowThresh}$ a jet coreline is allowed to remain under the wind speed threshold. When setting the value too high, lines might be found that are not actually jets. Setting $n_{predSteps}$, i.e., the number of predictor steps per iteration, too high leads to inaccurate results, as it takes the predictor too far away from the ridge line, making it more difficult for the corrector. A single predictor step is recommended. The parameter that influences the performance the most is $n_{corrSteps}$, the number of corrector steps performed in an iteration. When chosen too small, less lines merge, which increases

the computation time. Likewise a too large number, i.e., when converging onto ridge lines, causes computation overhead. The number of corrector iterations $n_{corrSteps}$ in the predictor-corrector procedure allows the user to balance how closely the jet stream follows a wind magnitude ridge line (high number) or the wind vector field (low number). In Fig. 6, we display the

**Table 1.** The extraction time in seconds when changing one parameter value while keeping the others fixed, here listed for the default parameters and two variations from the default parameters. Increasing $n_{stepsBelowThresh}$ prolongs the integration duration since longer lines emerge. Increasing $n_{predSteps}$ per iteration traces out the jet lines with less iterations. The smaller $\tau_{wind}$ (in ms$^{-1}$), the more lines are traced. With lower $n_{corrSteps}$ less jets merge causing higher tracing cost. For higher $n_{corrSteps}$ the runtime increases linearly due to more steps.

| Parameter | Parameter value | | | Extraction time [s] | | |
|---|---|---|---|---|---|---|
| | Default | Variation 1 | Variation 2 | Default | Variation 1 | Variation 2 |
| $\tau_{wind}$ | 40 | 30 | 50 | 10.1 | 12.2 | 8.2 |
| $n_{stepsBelowThresh}$ | 100 | 0 | 1500 | 10.1 | 9.6 | 11.6 |
| $n_{predSteps}$ | 1 | 2 | 5 | 10.1 | 8.8 | 9.8 |
| $n_{corrSteps}$ | 5 | 1 | 12 | 10.1 | 13.0 | 12.5 |

resulting jet corelines for different numbers of corrector steps, i.e., different degrees of regularlization. A cross-section of the velocity magnitude scalar field shows that with regularization the curves pass close to the ridge lines, but they result in smoother lines instead. The accompanying video contains animations of the cross section plane, showing that the regularized curves show reasonable agreement with the sectional extrema.

## 4 Interactive Visualization System

We implemented our integration-based jet coreline extraction in an interactive visualization system in order to visually analyze the jet streams in the context of warm conveyor belts and the tropopause. Our system is implemented using the Visualization Toolkit (VTK) (Schroeder et al., 2006).

### 4.1 Overview

The visual analysis of meteorological data is generally challenging, since the data are three-dimensional, time-dependent and contains a number of variables that interact with each other. Furthermore, numerical weather simulations are produced at increasingly higher resolution, resulting in several TB per case. Because the time series data are far too large to be held in memory, it is necessary to calculate features, such as the tropopause geometry, warm conveyor belt trajectories, and jet stream corelines once in a pre-process. The resulting geometric descriptions of the meteorological features are not only valuable for interactive visualization, they can also be input to traditional data analysis pipelines. To increase the compatibility with the accustomed workflows, we implemented the feature extraction in a command line tool, including the computation of the jet corelines, warm conveyor belts and the tropopause. The code is available in the supplemental material. Subsequently, the features are passed on to an interactive analysis tool, with which we explore jet stream corelines for multiple use cases:

- we show a case in which a warm conveyor belt outflow feeds a jet stream, thereby accelerating it;

- we demonstrate that our extracted jet streams align well with flanks of the tropopause;

- we visualize a setting in which the outflow of a WCB coincides with a displacement of a jet;

**Figure 7.** Our pipeline consists of two steps: a *preprocessing* that calculates the tropopause, WCB and jet coreline geometries using a command line tool, and an *interactive analysis* in which the user interactively explores the correlation of those features in 3D.

    – we display the tropopause at locations where jet streams split and merge.

To this end, the tool supports the combined visualization of features, the computation of derived properties, and interactive filtering operations. An overview of our pipeline is provided in Fig. 7. Next, we formally define the meteorological features and describe the components of the interactive visualization system.

### 4.2 Feature Extraction

Let $\mathbf{v}(\mathbf{x},t) : \mathbb{R}^3 \times \mathbb{R} \to \mathbb{R}^3$ be the time-dependent wind vector field. Its spatial coordinates $\mathbf{x} = (\phi, \lambda, p) \in \mathbb{R}^3$ are measured in latitude, longitude, and pressure. The wind velocity components are in $\mathrm{ms}^{-1}$ (horizontally) and $\mathrm{Pa\,s}^{-1}$ (vertically). The thresholds used in the definitions have been chosen empirically, and can be adjusted depending on the analysis task.

**Jet Stream Coreline**

The jet stream corelines are extracted by the algorithm described in Section 3. In the following, we denote jet corelines in the time-dependent horizontal wind magnitude field $s(\mathbf{x},t) = \|(u(\mathbf{x},t), v(\mathbf{x},t))\|$ as instantaneous curves $\mathbf{c}(\tau) : \mathbb{T} \to \mathbb{R}^3$, arclength-parameterized by $\tau \in \mathbb{T} \subset \mathbb{R}$. Since these lines evolve over time $t$, we introduce a time-dependent set $\mathcal{C}(t) = \{\mathbf{c}_i(\tau)\}$, which contains all jet corelines for a given time $t$, where $i$ is the index of the line.

**Tropopause**

We define the tropopause to be the largest connected surface composed of isosurfaces with isovalue $2\,\mathrm{pvu}$ and $-2\,\mathrm{pvu}$ of the potential vorticity field on the Northern and Southern Hemisphere, respectively. Other choices for the PV thresholds are imaginable, cf. (Highwood et al., 2000; Schoeberl, 2004; Kunz et al., 2011). Further, the pressure was required to be below a threshold of $740\,\mathrm{hPa}$ to remove false-positives near the ground. Note that this pressure threshold is by no means a general

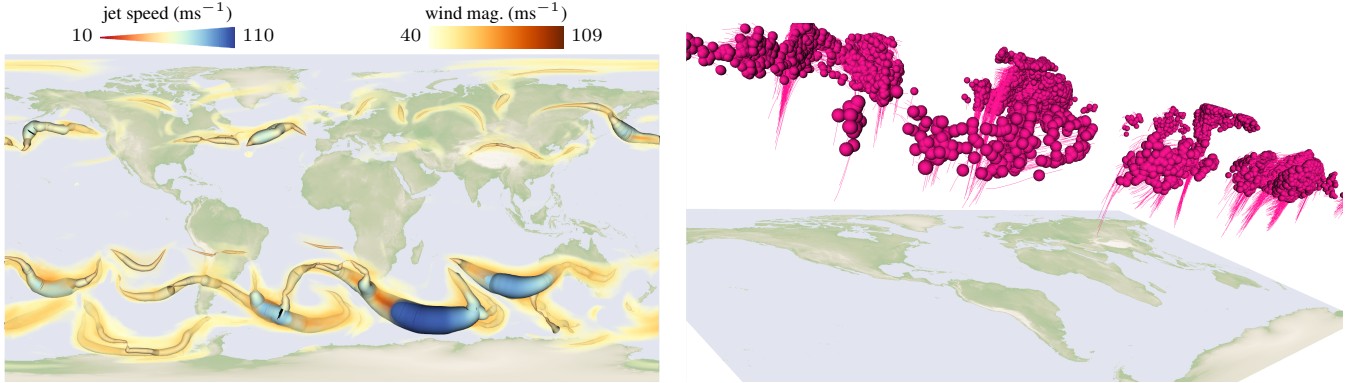

**(a)** *Jet stream with wind magnitude volume rendering*  **(b)** *Warm conveyor particles with pathlines*

**Figure 8.** Jet stream corelines are rendered as tubes, with jet speed being mapped to color and radius. Left: with volume rendering of wind speed, Right: warm conveyor belt particles are shown as particles with pathlines attached to convey motion.

constant. For example, Lillo et al. (2021) identified tropopause droppings well above the threshold at intense sub-synoptic scale events. Formally, the time-dependent tropopause is expressed as $\mathcal{T}(t) \subset \mathbb{R}^3$ with:

$$\mathcal{T}(t) = \{\mathbf{x} : P(\mathbf{x},t) = +2\,\mathrm{pvu} \wedge \phi > 0 \wedge p < 740\,\mathrm{hPa}\} \cup \{\mathbf{x} : P(\mathbf{x},t) = -2\,\mathrm{pvu} \wedge \phi < 0 \wedge p < 740\,\mathrm{hPa}\} \tag{6}$$

where $P(\mathbf{x},t)$ denotes the potential vorticity (PV) field. For procedures on how to address the sign flip at the equator, we refer to Schoeberl (2004); Manney et al. (2011) for a formulation based on an isentropic isosurface. In our work, we extracted the PV isosurfaces for the Northern and Southern Hemisphere separately.

**Warm Conveyor Belts**

Formally, we describe the set of WCB trajectories $\mathcal{W} = \{\mathbf{x}(t)\}$ as the union of all pathlines $\frac{\mathrm{d}\mathbf{x}(t)}{\mathrm{d}t} = \mathbf{v}(\mathbf{x}(t),t)$ of the wind

velocity field $\mathbf{v}(\mathbf{x},t)$ that ascend within $T \leq 48\,\mathrm{hrs}$ more than $\Delta_p = 600\,\mathrm{hPa}$:

$$\mathcal{W} = \left\{ \mathbf{x}(t) : \mathbf{x}(t) = \mathbf{x}_0 + \int_{t_0}^{t_0+T} \mathbf{v}(\mathbf{x}(t),t)\mathrm{d}t \ \wedge \ \left( \exists \tau \in [0,T] : p(t_0) - p(t_0+\tau) > \Delta_p \right) \right\} \tag{7}$$

To form $\mathcal{W}$, the trajectories are seeded from a dense space-time grid with initial coordinates $\{(\mathbf{x}_0, t_0)\}$ near the ground, using a horizontal grid spacing of $80\,\mathrm{km}$, 14 equidistant vertical levels between 1050 and $790\,\mathrm{hPa}$, and a temporal spacing of $6\,\mathrm{hrs}$. Trajectories are numerically integrated using a fourth-order Runge-Kutta integrator.

**Combining Eulerian and Lagrangian Features**

Note that the tropopause surface $\mathcal{T}(t)$ and the set of jet stream corelines $\mathcal{C}(t)$ depend on the observation time $t$. This means, they are Eulerian features, i.e., they are computed per time slice $t$. The WCB trajectories in $\mathcal{W}$, on the other hand, exist over multiple time steps, i.e., these are Lagrangian features. Care must be taken when visualizing Eulerian and Lagrangian features together in

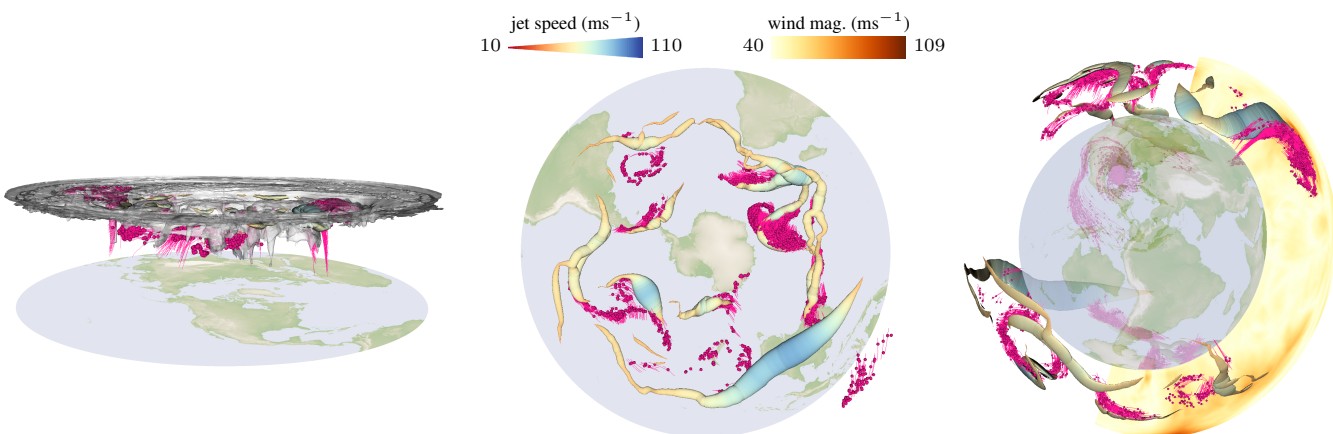

**Figure 9.** Interactive visualizations of acceleration, movement, and splitting and merging of jet streams. Here, polar stereographic projections of the tropopause, jet streams and warm conveyor belt (WCB) outflows (magenta) for the Northern Hemisphere (left), Southern Hemisphere (middle), and a 3D globe view with interactive volume slices (right) are shown. Jet speed is mapped to tube color and tube radius.

space, since points on these two structures might be far apart in time, potentially leading to wrong conclusions. Since the dense
set of WCB trajectories is furthermore prone to generate visual clutter, we extract the locations of WCB trajectories that exist at
a certain time $t$, which results in a set of points $\mathcal{W}(t)$:

$$\mathcal{W}(t) = \{\mathbf{x}(\bar{t}) \in \mathcal{W} : \bar{t} = t\} \tag{8}$$

This point set contains all locations that are reached by any of the WCB trajectories at a certain time $t$, regardless of their seed
time. Since the complete set of WCB points at a certain time $t$ is still too dense, and since we are primarily interest in the
interaction of WCBs and jets, we filter this point set based on the proximity to a jet in Eq. (9). For this, we empirically use a
horizontal distance threshold of $15°$ latitude-longitude and a vertical pressure difference of at most $50\,\mathrm{hPa}$:

$$\mathcal{W}^*(t) = \{\mathbf{x}(\bar{t}) \in \mathcal{W} : \bar{t} = t \;\wedge\; (\exists \mathbf{c}(\tau) \in \mathcal{C}(t) : \|\mathbf{c}(\tau) - \mathbf{x}(t)\| < 15° \;\wedge\; |p(\mathbf{c}(\tau)) - p(\mathbf{x}(t))| < 50\,\mathrm{hPa})\} \tag{9}$$

### 4.3 Visualization

**Visual Mapping**

Rautenhaus et al. (2015b, a) developed seminal 3D visualization methods for the analysis of jets, WCBs, and the tropopause
using the framework Met.3D. In the following, we describe how we visualize jets, WCBs, and the tropopause, for which we
similarly use a combination of particles, tubes and isosurfaces. The tropopause $\mathcal{T}(t)$ is extracted as isosurface of the potential
vorticity field and can be rendered semi-transparently in order to allow a view onto the structures behind it. To encode the
velocity magnitude of jets $\mathcal{C}(t)$ and to convey a 3D impression of their shape, we map the magnitude to both color and tube
radius using transfer functions, i.e., color and tube radius are dependent on the magnitude. Color bars above the visualizations

depict the mapping throughout the paper. The mapping to the radius is adjusted to align the resulting radius with the actual spatial extent of the jet streams. The warm conveyor belt particles $\mathcal{W}^*(t)$ at the selected time slice are rendered as magenta spheres. To encode temporal information about their motion, we display short pathlines in both forward and backward time, originating from the warm conveyor belt particles, see Fig. 8b. To provide context, we add a world map with image courtesy by
Reto Stockli, NASA Earth Observatory Group.

### Viewing Projections

While viewing the scene with an equirectangular mapping has many advantages, it can be beneficial to view the scene in a different projection. For example, if we want to examine parts of the jet stream that moved towards the poles. In this case, viewing the scene on a polar stereographic map gives us a more accurate picture because as soon as the jet comes close to the
poles on the equirectangular map, it becomes one long tube due to distortion. However, this projection can only show either the Northern or the Southern Hemisphere. In addition, we provide the user with the option to view the data on a 3D globe, conveying a correct depiction of sizes. The different projections are shown in Fig. 9. Alternatively, any other projection such as a north polar orthographic or a north polar Lambert equal area projection would be imaginable as well.

### User Interaction

In all examples, the vertical axis (distance to ground) is scalable by the user. Further, the time slider, transfer functions of color and radius, as well as the filtering thresholds can be interactively adjusted to explore their effect. To provide further context, image slices, isosurfaces and direct volume renderings of additional scalar fields can be added and interactively adjusted. Fig. 9 (right) gives an example of a slice showing the velocity magnitude.

## 5    Applications

In the following, we visualize jet stream corelines in the context of warm conveyor belts and the tropopause. The close alignment of jet corelines with folds of the tropopause, is a strong indicator for the plausibility of the extracted jets Maher et al. (2019). Further, the influence of WCBs is reflected in the behavior of the jet in the case studies. We refer to the supplemental material for time series animations.

### Jet Acceleration

In Fig. 10, we visualize how a WCB outflow that intersects with a jet stream can greatly accelerate it. In Fig. 10a, we see how a WCB approaches the jet stream and in Fig. 10b, we see the situation two days later. We can observe a considerable acceleration of the wind speed at the jet stream core. There are several reasons why these local jet maxima, so-called jet streaks, are of eminent importance in dynamic meteorology: (i) the entrance and exit regions of jet streaks are associated with an ageostrophic (vertical) circulation, which – for instance – favors lifting motion in the right exit and thus is conducive for storm development
and the initiation of convection (Shapiro and Keyser, 1990); (ii) jet streaks have also been identified in upper-tropospheric

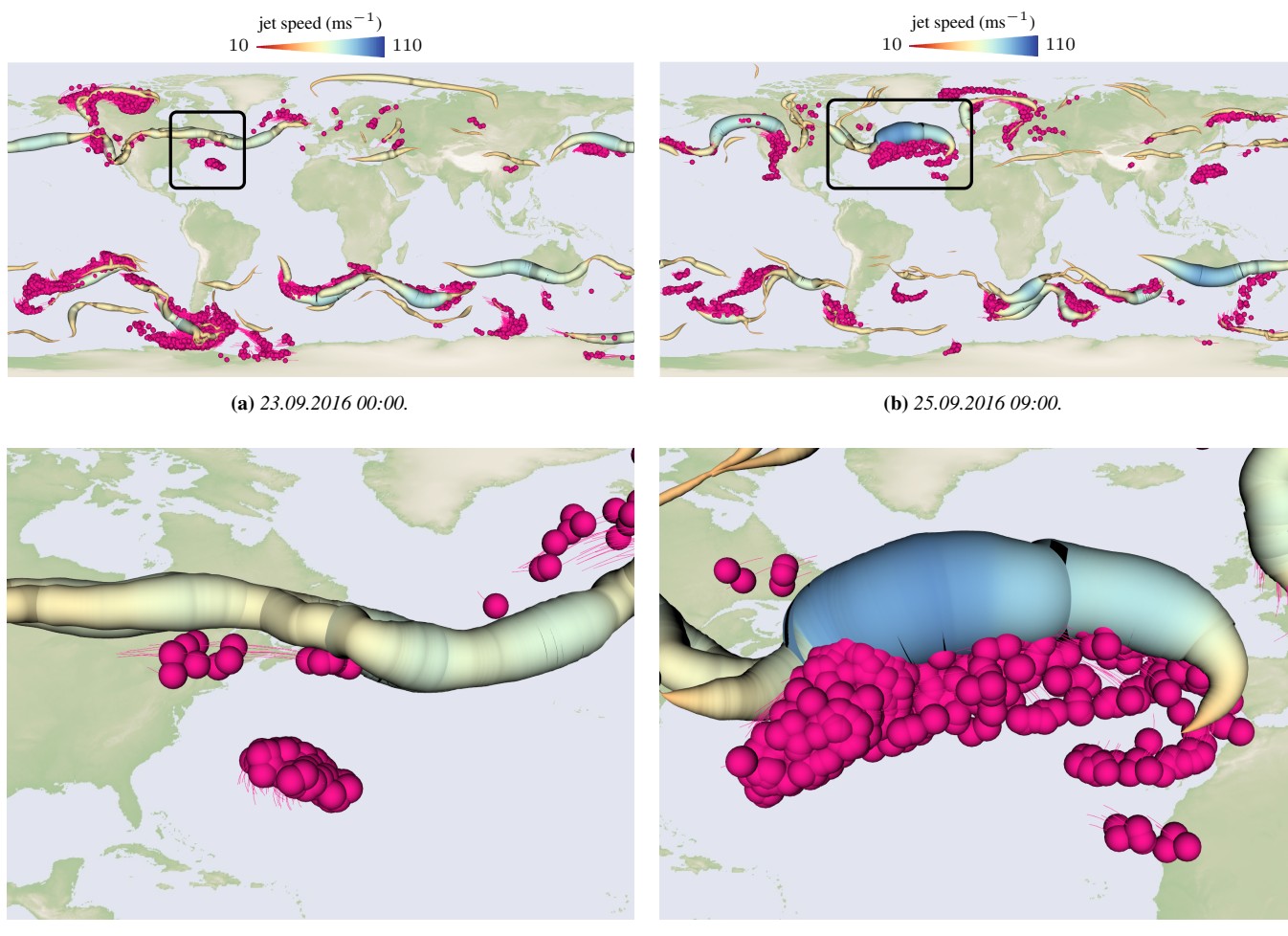

**(a)** *23.09.2016 00:00.*

**(b)** *25.09.2016 09:00.*

**(c)** *close-up of (a), 23.09.2016 00:00.*

**(d)** *close-up of (b), 25.09.2016 09:00.*

**Figure 10.** Visualization of jet streams $\mathcal{C}(t)$ and WCB particles $\mathcal{W}^*(t)$. (a) shows a WCB outflow moving northwards in direction of the jet stream. (b) shows the situation two days later. The WCB approached the jet stream and a considerable acceleration of the wind speed at the jet core is observable (dark blue jet). For both images, close-ups are shown in (c) and (d). Jet speed is mapped to tube color and tube radius.

regions with substantial cross-tropopause mass fluxes, i.e. ozone-rich stratospheric air might be transported because of them from the stratosphere into the troposphere (Sprenger et al., 2003); and (iii) the jet streaks themselves have a distinct lifecycle that can, after their genesis, influence the flow evolution and the weather predictability far downstream of the genesis region (Cunningham and Keyser, 2002). Jet-accelerating interactions are currently a particularly active topic in WCB research. In fact, Oertel et al. (2020) and Blanchard et al. (2021) recently identified negative-PV structures produced in convective ascents embedded in a WCB that then locally accelerated the jet.

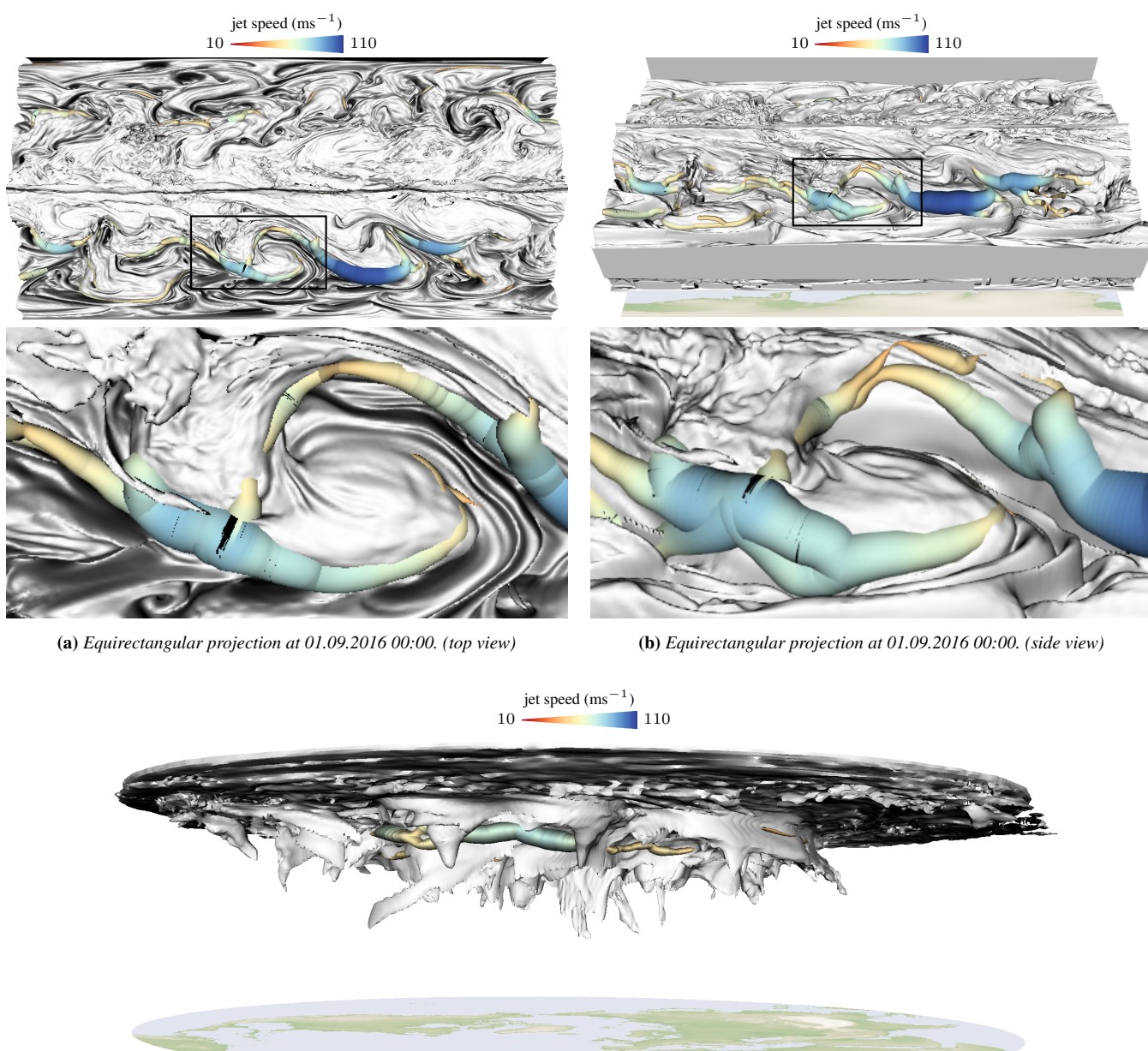

**(a)** *Equirectangular projection at 01.09.2016 00:00. (top view)*

**(b)** *Equirectangular projection at 01.09.2016 00:00. (side view)*

**(c)** *Polar stereographic projection at 03.09.2016 18:00.*

**Figure 11.** Interaction of jet stream and tropopause. The jet stream is located where the tropopause is steep. Jet speed is mapped to tube color and tube radius.

### Jets at the Tropopause

According to Winters et al. (2020); Maher et al. (2019); Koch et al. (2006), the jet stream is dynamically linked to locations with a strong potential vorticity gradient. By the dynamic definition of the tropopause, this is where the extracted tropopause

isosurface is steep. Winters et al. (2020); Maher et al. (2019); Manney et al. (2014) showed that extratropical westerly jet cores are located along the flanks of 'valleys' or folds of the dynamic tropopause, i.e., where the tropopause shows substantial (vertical) excursions from a smooth and horizontal basic state. Fig. 11 shows jet streams and the tropopause geometry using an equirectangular and polar stereographic projection. Especially in the supplemental video, the connection between jet stream and tropopause is apparent. As the tropopause evolves over time, the jet stream changes accordingly, and a smooth temporal evolution of the jets at the flank of the tropopause is a useful plausibility measure. The three-dimensional visualizations reveal not only the complicated 3D structure of the tropopause, they also illustrate the jet stream location relative to the folds at multiple pressure levels. This allows the complex co-evolution of the two features to be studied, which would be difficult to achieve in *simple isentropic or isobaric 2D* visualizations. A 3D view of the co-evolution may prompt further hypotheses to be investigated by means of a dynamical analysis of the involved physical processes, which might shed further light on the mechanisms governing the downstream propagation, and potential amplification, of jet (or PV) anomalies, with significant implication for weather predictability (Grams et al., 2011).

**Jet Displacement**

Because the jet stream is dynamically linked to locations with a strong potential vorticity gradient (Koch et al., 2006), we now visualize WCBs, PV at 270 hPa and the jet stream. In Fig. 12a, we see a WCB approaching the jet stream, and in Fig. 12b, we observe the situation two days later. We observe that a displacement of the jet stream occurs in the presence of the WCB as Steinfeld et al. (2020) found while examining the same case as shown in Fig. 12. As discussed before, WCBs can influence the jet in different ways: either they lead to local wind speed maxima (jet streaks, as in Fig. 10), or they directly displace the jet from its initial position (Oertel et al., 2020; Blanchard et al., 2021; Joos and Forbes, 2016). Often, this jet perturbation is considered in a PV perspective. This is so because WCBs transport low-PV air into upper-troposheric levels, which in turn can help to enhance or redirect local PV gradients and jets. By considering the jet as a 3D feature (instead of a PV gradient on a single isentropic or isobaric level), the interaction of the WCB with and the impact on the wind speed becomes more direct. A better understanding of jet displacements and accelerations by WCBs is of interest to the NWP community, because of the high relevance for weather predictability downstream of the WCB/jet interaction (Grams et al., 2011; Rodwell et al., 2018). In fact, Davies (2015) used the term 'weather chains' to highlight that upstream impacts, e.g., WCB/jet interactions in the North Atlantic, influence the weather downstream, e.g., over Europe. Currently, it is discussed whether WCBs are partly responsible for so-called forecast busts (Rodwell et al., 2013), i.e., particularly poor forecasts.

**Jet Split and Merge**

Jet streams are rarely one connected band around the planet. The jet streams can split and merge every now and then and even the polar and the subtropical jet can merge to one single jet stream (Ahrens and Henson, 2018; Breeden et al., 2021). The tropopause might exhibit an interesting three-dimensional structure at those events. Thus, in Fig. 13, the automatically extracted split and merge events are shown, where split events are encoded with green spheres and merge events with purple spheres. In the images below, a scenario is shown in which jets split or merge depicting how their behavior is influenced by the three-dimensional shape

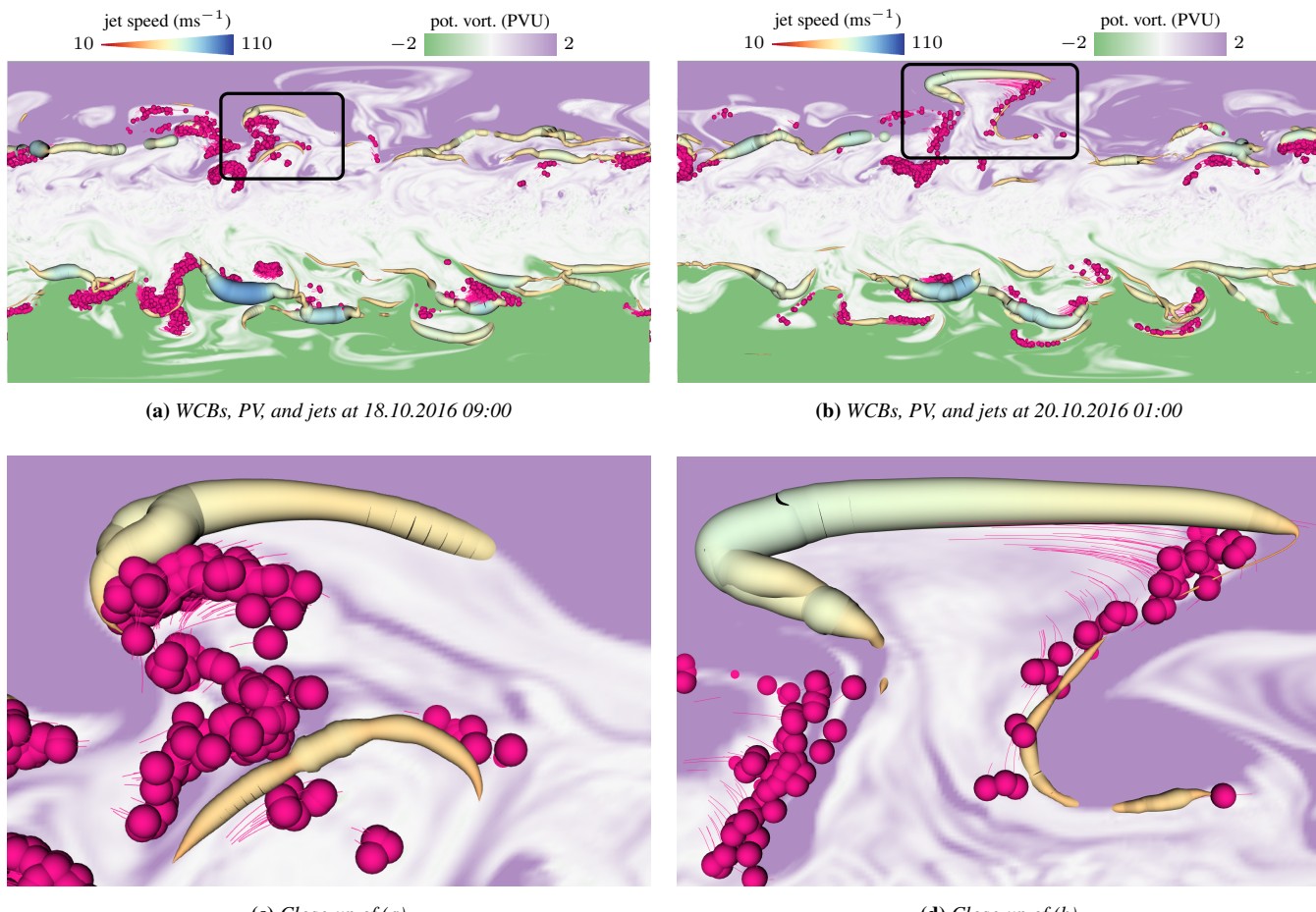

**(a)** *WCBs, PV, and jets at 18.10.2016 09:00*

**(b)** *WCBs, PV, and jets at 20.10.2016 01:00*

**(c)** *Close-up of (a)*

**(d)** *Close-up of (b)*

**Figure 12.** Visualization of jet stream and WCBs together with the potential vorticity at the upper troposphere at 270 hPa. Green values indicate PV smaller than −2 pvu and violet values indicate a PV larger than 2 pvu. White values are close to 0 pvu. On the jet stream, wind speed is mapped to color and tube radius. (a) shows how a WCB pushes against the jet. (b) shows the situation two days later. The WCB injected low-PV air masses into the upper troposphere and pushed the jet stream northwards. In (c) and (d), close-ups are shown.

of the tropopause. Some examples of single-jet and/or double-jet configurations are well known from the North Pacific and North Atlantic, where wave activity can be transferred from one to the other jet (Martius et al., 2010), which can sometimes lead to the merging and/or splitting of jets. In addition to the ocean-wide implication of jet splittings and mergings, it is conceivable – yet still not systematically studied – that splitting and merging events are meteorologically 'active' regions, for instance with enhanced probabilities for clear air turbulence (CAT) and thus impact on aircraft efficiency and safety (Reiter and Nania, 1964). It will be a rewarding aim to further study these flow regions, not only with respect to CAT but also other meteorological impacts, e.g., stratosphere-troposphere exchange (Akritidis et al., 2016).

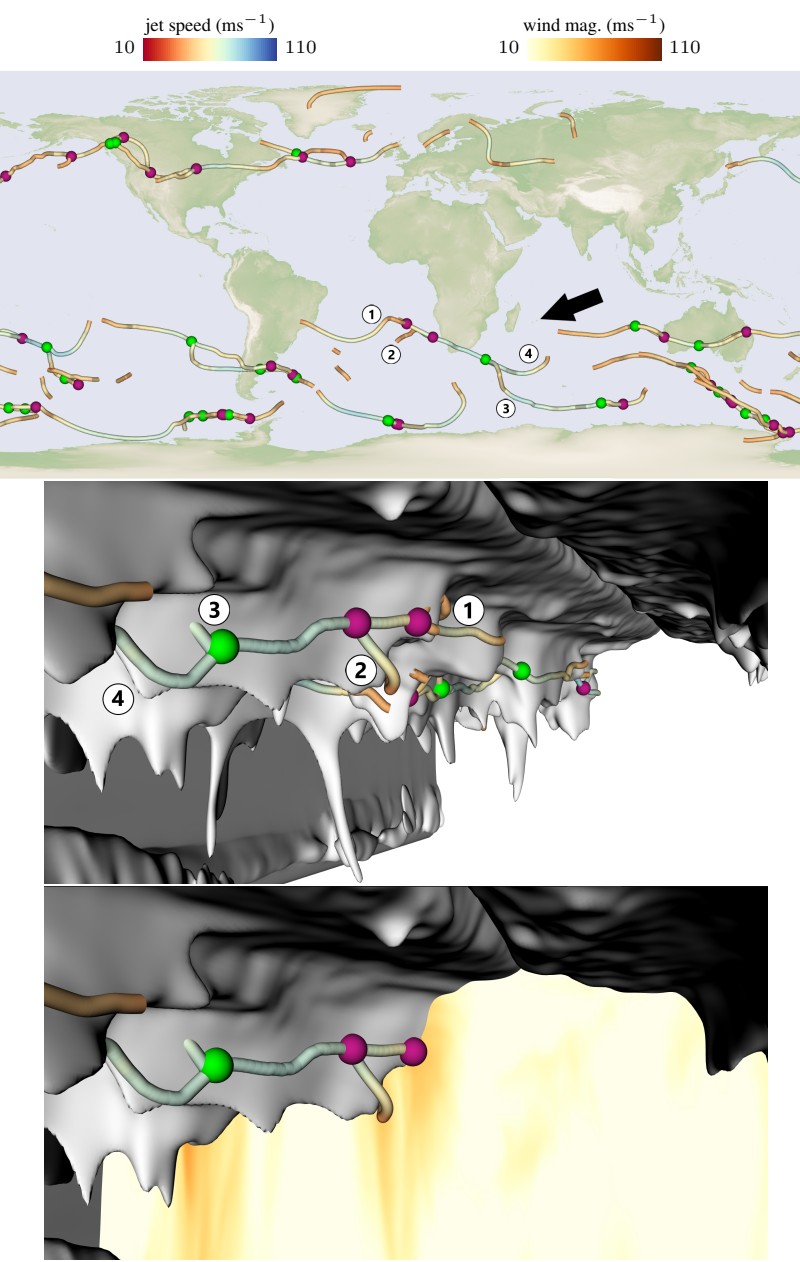

**Figure 13.** Top: Overview visualization of split (green) and merge (purple) events of jet stream corelines at the tropopause. An arrow depicts the camera position of the 3D views shown from below the tropopause. Middle: A jet (1) moves along a tropopause fold and another jet that passed from below the tropopause fold merges into it (2). The combined jet later splits into two separate branches (3) and (4). Bottom: an orthographic slice color-codes the wind magnitude, showing that the jets pass through orange regions (high wind speed). Here, for 09.09.2016 12:00.

## 6 Conclusions

Jet streams are three-dimensional meteorological flow features that can interact with other atmospheric structures such as warm conveyor belts and the tropopause. We developed an integration-based feature extraction algorithm that locates the explicit jet stream geometry near the tropopause. In contrast to previous local methods (Kern et al., 2017), the predictor-corrector approach allows the wind magnitude ridge lines to be regularized. When the ridge line exhibits high vertical curvature, then regularizing with a smooth vector field helps to produce smoother lines. The high vertical curvature of the ridge line is a product of the low vertical resolution of the hybrid model levels. With an increased model resolution, such regularization will hopefully not be necessary anymore in the future. Apart from this, local feature extractors such as parallel vectors often experience fragmentation independent of the discretization of the domain, resulting in spurious lines that have to be reconnected in a post-process. Predictor-corrector approaches are in the class of integration-based methods, which generally avoid this numerical issue. Given long and coherent jet stream trajectories, we can automatically detect proximities to WCBs, and locate split and merge events of jets. To visualize jets along with WCBs and the dynamic tropopause, we developed a visualization system and applied the tool in multiple cases, visualizing the acceleration and displacement of jets near WCBs, and the movement and split/merge along tropopause folds. It is known that WCB outflows significantly influence the upper-level waveguide, which is typically identified as a region of enhanced PV gradients on an isentropic surface (Grams et al., 2018; Spreitzer, 2020; Saffin et al., 2021; Grams et al., 2011). However, this perspective does neglect a crucial aspect of the atmospheric flow setting: its three-dimensionality. The distance between the WCB outflow and the waveguide has to be determined. This, however, is no trivial task. It becomes more tangible if the interaction is not only considered in the PV perspective, but instead the WCB outflow trajectories are directly linked to the jet corelines. The feature extraction (WCB, jet corelines, tropopause) developed in this study gives the means to visually inspect co-occurences of WCB outflows and the jet corelines with respect to their horizontal and vertical position, with the amplitudes (local wind maxima; jet streaks) and their overall geometric structure.

In the future, it would be interesting to extract jet streams automatically for even longer time series. To increase temporal coherence, multiple time steps could be taken into account. Since our extraction traces one jet coreline after the other, varying the order in which the seed points are processed influences the jet coreline network, which becomes noticeable when too little weight was given to the corrector step, i.e., when the lines primarily follow the flow rather than the ridge lines. It would be interesting to investigate, how the jets extraction could be made order-independent. While the extraction algorithm could be applied to data arising in an operational context, more work is necessary for a successful integration in operational routines, including an increased temporal stability, heuristics for automatic parameter selection, and a requirement analysis with operational forecasters to integrate additional constraints into the feature definition and extraction.

*Code and data availability.* A C++ implementation of the integration-based jet stream core extraction is available at https://doi.org/10.5281/zenodo.5567863 and on GitHub at https://github.com/fau-vc/jet-core-extraction. The demo data is published at https://doi.org/10.5281/zenodo.5567866. ERA5 data are available from the Copernicus Climate Change Service at ECMWF.

*Video supplement.* A supplemental video shows the extraction results in an animation. The video is made available at https://doi.org/10.5281/zenodo.5722311.

*Author contributions.* L.B. implemented the method, conducted the analysis and wrote the manuscript, M.S, M.B., H.J., T.G. conceived the
idea and M.S, M.B., H.J. provided the data. All, M.S, M.B., H.J. and T.G. have supervised the work with regular inputs and have contributed to the writing of the manuscript.

*Competing interests.* The authors declare that the research was conducted in the absence of any commercial or financial relationships that could be construed as a potential conflict of interest.

*Acknowledgements.* Tobias Günther was supported by the Swiss National Science Foundation (SNSF) Ambizione grant no. PZ00P2_180114.
Maxi Boettcher acknowledges funding by the European Research Council 485 (ERC) under the European Union's Horizon 2020 research and innovation programme (project INTEXseas, grant agreement no. 787652).

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
