# Peer review of "Integration-based Extraction and Visualization of Jet Stream Cores"

_Geoscientific Model Development, 2021_

## Referee Comment (RC2)

**Review of "Integration-based Extraction and Visualization of Jet Stream Cores", by Bösiger et al.**
(Review by Gloria Manney)

**Recommendation:** Publish after some clarifications and minor revisions.

**General Comments:** This paper presents a new method and software for tracking and visualizing jet stream cores. This is a potentially very useful new method with some important advantages and should be a valuable addition to existing tools for jet characterization and analysis. As such, it should be appropriate for publication in GMD and I would expect there to be much interest in it among atmospheric scientists who focus on studies of the jet stream (including myself!). However, I feel there are some important changes to the presentation needed to (1) better reflect previous work on jet stream characterization and the phenomena (WCBs, tropopause structure) the jet streams are related to here, and (2) to make the paper more accessible to an audience of atmospheric scientists for whom this method / software may be very useful but who may not in general be computer scientists or mathematicians. (I believe I'm a reasonable example of this class of atmospheric scientist, so that if I don't understand some things it is not unlikely that many other interested readers will be in the same position.) These changes are summarized here (with some further specific examples given in the "specific/minor comments" below):

(1) A few terms and some notation are used throughout this paper that are not (clearly) defined or are things many in your audience may not be familiar with, and should be defined and/or expressed in plainer language:
   (a) Voxel -- should just be defined the first time it is used as it will be unfamiliar to many readers (as I understand it from looking up the definition, it is nothing more than the 3-D analog of a pixel).
   (b) Heuristic (heuristics, heuristically) -- in general (and in many fields) this term is often (perhaps over) used and frequently not clearly defined (thus sometimes mis-used). Indeed, dictionary definitions are many and varied. My impression is that the way you use it here is something akin to "pertaining to a trial-and-error method of problem solving used when an algorithmic approach is impractical", or to simply say that the process in question requires human intervention (e.g., the necessity to make choices based on things the human eye does very well but we tend to have trouble telling computers how to do). A more specific statement (and perhaps examples from some of the previous studies you cite) of what you mean by "heuristic" would be very helpful in motivating the development and advantages of your method.
   (c) Manifolds -- I question the need to use this term (which readers unschooled in topology may not recognize or may immediately assume is expressing some complicated concept) when the fundamental information conveyed in this context by "instantaneous 1-manifolds" is that it is a line/curve (1-dimensional) at a particular time, and by "time-dependent 2-manifold" that it is a time-varying surface (2-dimensional).

(d) Several definitions (tropopause, WCBs, and "filtered" WCBs) used are expressed in set-builder notation (which many readers may not be familiar with); in general, you explain these (though not always completely) in words beforehand, but it isn't always obvious that that is what you are doing. I'd suggest that you make this relationship explicit, by saying something like (e.g., for the tropopause case): "...largest connected surface of 2 PVU (-2 PVU) in the northern hemisphere (southern hemisphere) at pressures below 740 hPa, that is: " (in fact, since this is a very common way to define the tropopause, any atmospheric scientist interested in your work will immediately understand this without the equation, so it is not obvious that you need the equation at all -- however, I have no problem with including it as long as it is also explained in plain words so that the reader will understand regardless of whether they are familiar with the notation.

(2) The citation of previous literature is lacking, especially with regard to upper tropospheric jet streams and characterisation thereof. While I understand that the focus here is on the software tools developed, these are being presented specifically in relation to jets in the Earth's atmosphere and thus the primary audience is atmospheric scientists -- hence it is important to accurately relate this work to previous work in the field and to the reasons why the method may facilitate future work. In particular:

    (a) Section 2.2: There are much better references than Dameris (2015) for what the tropopause is and why it is important; in addition, Dameris (2015) is not readily publicly available. I would start with the reviews by Holton et al (1995, Rev Geophys) and Stohl et al (2003, JGR). In addition to Skerlak et al (2015), I would add a couple of classic papers for tropopause structure such as (cited by Skerlak et al) Danielsen (1968) and Shapiro (1980) -- or at least add "and references therein". Highwood et al (2000, QJRMS), Schoeberl (2004, JGR), and Kunz et al (2011, JGR) are good references for the range of PV values that have been used for dynamical tropopause identification and for what regions / purposes different values are appropriate.

    (b) Section 2.3: This is a very incomplete and biased discussion of upper tropospheric jets and previous work characterizing them.

        (i) Ahrens and Henson (2018) is not readily publicly available, and there are numerous choices for classic work describing the jet streams and their importance to the atmospheric circulation. Koch et al (2006) and Schiemann et al (2009) (already cited in this preprint) both give concise historical introductions. Harnik et al. (2016) provide a nice brief review in relation to jet regimes and extreme weather events.. Manney et al (2014, J Clim) and Manney and Hegglin (2018, J Clim) have in their introductions comprehensive discussions of the literature in the context of the importance of and variations in jet streams, which provides many of those classic references.

        (ii) Several methods for identifying and characterizing jet streams that seem very relevant to this work are not mentioned, including the method

introduced by Manney et al (2011, ACP) and used in Manney et al (2014, 2021, J Clim) and Manney and Hegglin (2018) (with more physically-based distinctions of the subtropical and polar jets in the latter two of those papers); the method used by Winters et al (2020, MWR, and several references therein); and that of Maher et al (2020, Clim Dyn). Should also cite Spensberger & Spengler (2020, J Clim) in addition to Spensberger et al (2017) and note that their method does the characterization on the dynamical tropopause.

(iii) There are many jet characterization methods / studies in which the "assumption that the flow is oriented eastwards" is not made (and others where it is only used after the fact), including Manney et al (2011 ACP,) (method also used by Manney et al (2014) and refined for Manney and Hegglin (2018) and Manney et al (2021).) In addition several methods (including that of Manney et al, above references) do characterize the jet position/extent in the vertical as opposed at a level or in a layer.

(iv) The statement that the jets "can be further classified into different types based on their location" is vastly oversimplified, and, given the usage of these terms later in the paper, the physical distinction between subtropical and polar jets should be discussed accurately here, as well as the fact that there is indeed a spectrum of jets that may have characteristics that are a hybrid between the two (see Lee & Kim, 2003, JAS; Manney et al, 2014, 2021; Winters et al, 2020; and references therein). Peña-Ortiz et al (2013), in fact, noted that attempting to distinguish polar and subtropical jets by latitude was commonly unsuccessful; Manney et al (2011, 2014) noted that using a simple latitude criterion was only useful for very broad climatological studies, and Manney and Hegglin (2018) introduced a more physically-based method of distinguishing subtropical and polar jets based on tropopause height changes across the jet region. Winters et al (2020, and references therein) distinguish subtropical and polar jets by identifying them in different isentropic layers, and show clear instances of them merging to form a jet with hybrid characteristics.

(c) I am not as familiar with the literature on WCBs, but the discussion strikes me as often making general statements without giving citations (some instances noted below in the specific comments).

(3) General Questions (those with ** at the beginning are more further information / general interest questions, rather than necessary changes to this manuscript):

(a) Rationale for choices of thresholds/definitions, and discussion of sensitivity to those thresholds/definitions (there is a statement on line 279 that "thresholds used in the definitions have been chosen based on common practices in atmospheric science", but you need to give references and briefly note the physical basis for that "common practice", including:

(i) 40 m/s for the minimum windspeed for the seed points. (Some information on the sensitivity of the performance of the software to this is given, but

no rationale is given for the default choice, nor is the sensitivity of the physical results to this discussed.)

(ii) 190 to 350 hPa for the domain for jet extraction. 190 hPa is not low enough to exclude the stratospheric "subvortex" jet, which commonly extends down to between 150 and 250 hPa (eg, 340K), especially in the SH late winter and spring (e.g., Manney et al., 2014). Manney et al (2014, their Fig. 6) fairly commonly identified jet cores of over 40m/s as low as about 5 km, near or at their high pressure search boundary of 400 hPa, as well as jet cores (distinct from the stratospheric subvortex jet, which they characterized separately) near 13--14 km (typically for subtropical jets at latitudes equatorward of about 30 degrees), very near or at their low pressure search boundary of 100hPa; as they noted (and identified), the stratospheric subvortex jet often overlaps considerably in altitude with the upper tropospheric jets.

(iii) 2 PVU for the dynamical tropopause (many other values are used, and higher values of 3 to 5 PVU have often been recommended for mid to high latitude features, eg, Highwood et al, 2000, QJRMS; Schoeberl, 2004, JGR; Kunz et al, 2011, JGR), and 740 hPa for the maximum pressure (e.g., intense sub-synoptic scale events can be associated with the tropopause dropping well above this pressure, eg, Lillo et al, 2021, JAS, and references therein). Also, if the domain studied extends into the tropics, how is the tropopause computed there since PV goes to zero (the most common procedure is to use an isentropic surface, commonly 380K, wherever the magnitude of the PV is less than the threshold above this isentropic level, eg, Schoeberl, 2004; Manney et al, 2011; and references therein)?

(iv) Thresholds for proximity of WCBs to jet coreline.

(b) I find the discussion overall somewhat unclear in the usage of "steps" -- there are spatial steps (eg, the grid spacing used for the prediction step), time steps, and procedural steps (eg, prediction and correction steps) and it is not always clear from the context which is being discussed.

(c) The representation of the corelines (which, as I understand it, are simply that, that is lines approximately connecting the core locations) as tubes, with wider tubes for higher windspeeds has the potential to confuse the reader into believing they show the jet region (analogous to the "regions" discussed in Koch et al, 2006 and Manney et al, 2011). While there will be some information since regions with higher windspeeds will have windspeeds above the threshold(s) over a larger area, there is by no means a direct correspondence since the wind gradients are not uniform or symmetrical around the core. The text needs to be very clear about this point, so as to not mislead the reader into thinking they are seeing the physical region where a jet is defined.

(d) While the terms subtropical and polar jets are tossed around in the paper there is apparently no attempt to distinguish these in a physically meaningful way. Thus statements suggesting that a jet coreline represents a subtropical or polar jet

should not be made.  **Also, it would be interesting to know if there are plans to add such a distinction to the method.

(e) Use of pressure rather than potential temperature for the vertical coordinate: Why was the pressure coordinate chosen for the jet extraction?  Given that an isentropic coordinate would be more "flow-following" on short (days to a week or two) timescales, would one expect substantial differences if the procedure were implemented in an isentropic coordinate?  **Would it be feasible to implement it in isentropic coordinates?

(f) **I would be interested in some more discussion (perhaps largely in an appendix or in the supplemental material) on the performance.  The description given is all per time step (and it is not entirely clear what the time step being referred to is).  Your study period is two months.  What is the total time to process that period?  The description of the procedure sounds storage-intensive -- what is the total storage needed for output for your study period?  What do the performance results imply about the feasibility of using this procedure for climatological studies?  From all of this, can you say something about the system requirements (CPUs/speed, memory, cache, storage) for running this effectively?

(g) **I would also be interested (again, in an appendix or supplementary material) in more specifics about the algorithms used for various steps.  I think such information could be very helpful to the reader who might want to implement something similar to parts of this but is not conversant with C++.

**Specific / Minor Comments (in order of appearance):**

Line 22, please provide (a) more accessible and foundational reference(s) per general comment (2).

Line 28--29, why is the tropopause expected to show highly 3-d structures around split and merge events?  Please give references for this.

Line 76--80, would be good to note somewhere in here that the tropopause altitude is generally highest in the tropics, lowest near the poles, and drops sharply across the subtropical jet.  It not uncommonly extends below 6km in folds or other tropopause depressions (see general comment (2a)).

Line 105, this is presumably a right-handed coordinate system, and v is defined as positive if northward?

Lines 108--112 (through eq. 3), this sentence / equation aren't very clear.  I'm guessing that the text is supposed to be a description of the following equation, but I don't know what the || means in this context (where it looks like an operator or something stating a relationship) nor whether the equation is supposed to represent the rephrasing of the problem or something related to the solver.  Please re-word.

Figure 1 caption, please clarify that the date/time at the end of the caption is that shown in the figure.

Lines 121-122, some general reference(s) should be given for cyclones.

Line 192, please explain what is meant by easing "the balancing between prediction and corrections steps" and why normalization accomplishes this.

Line 214, and Figure 2 caption.  If the weak endings are removed, why are there still green segments in Fig. 2(d)?

Line 292, what is the method for integrating dx(t)/dt for the trajectories?

Lines 316-317 & 331, I don't know what you mean by "transfer functions", please explain or reword (since it sounds like you are just saying both the radius and color are dependent on magnitude, you could simply say that).  It would be helpful to have some sort of a key for the radius on the plots; if the radius relationship is also linear (as the color one appears to be) you might make the color bar a wedge rather than a rectangle.  If the radius change is not linear with windspeed, you need to say that.  Related to this, and Figs. 10, 11, and 12, it needs to be explicitly stated that the radius does not show the region wherein a jet is defined, per general comment (3c).

Line 318, how did you determine that it was a stratospheric jet?

Line 326, why not use a north polar orthographic or a north polar Lambert equal area projection?  These emphasize the mid to high latitude regions more than the stereographic.

Figures 2, 8, 9 10, and 12, the color bars are too small.  Also, the choice of a diverging color palette for the windspeed in these and other figures seems a poor one, since it is a positive definite quantity for which it does not appear that there is a reason to emphasize a transition at one particular value -- a perceptually uniform sequential palette would be preferable.

Lines 340-343, per general comment (3d), you have not done anything to identify polar vs subtropical jets, which have different primary driving mechanisms and thus different characteristics (e.g., Lee & Kim, 2003; Manney et al.,2014; Winters & Martin, 2020; and references therein).  There are not "generally" two jets, in fact the patterns of jets and how many there are (with one to three being most common, but more possible at a given time/longitude) vary strongly with region and season (e.g., Manney et al., 2014, 2021, especially see Fig. 1 in the latter).  If you are going to use the terms, you need to provide some justification for referring to a particular jet as polar or subtropical since there are important physical distinctions between the two (and of course, some jets may have hybrid characteristics between the two).

Line 355--363, it would also be good to cite Winters & Martin (2020) and Maher et al (2020) (and references therein) here, since the methods they use (unlike Koch's) rely on those strong PV gradients.  There is a large body of work (much of it cited in these recent papers; also see Manney et al, 2014, and references therein) showing that extratropical westerly jet cores lie near/at the dynamical tropopause in the region where there are rapid altitude decreases in the tropopause altitude with increasing latitude, thus the "expectation" of it lying on the flanks of valleys in the tropopause, and of tropopause folds "wrapping" along the flank of a jet are well-known results, as is the complex 3D structure of the tropopause.   Per general comment (2b), there are many papers (including those cited previously in this review) that characterize the jet structure in both the horizontal and vertical, so the largely 2D views described in lines 362--363 are by no means "typical" and have not been for on the order of the last decade.

Lines 366--367, can you say anything about how the visualization (which, though informative and interesting, is qualitative) will help shed light on mechanisms.

Line 371, why 270 hPa?   Also, why on an isobaric rather than an isentropic surface?

Lines 372 & 378, is this really entirely an effect of the WCB on the jet?  That is, is there no effect of changes in the jet on the WCB?  How do you know which is causing which to change?

Lines 373--375, some references for these effects are needed.

Lines 385--386, it would be appropriate to cite Manney et al (2014, 2021), Homeyer & Bowman (2013, JAS), Winters & Martin (2020), Spensberger & Spengler (2020), and references therein here, per general comment (2b).

Lines 404--407, please provide some references for these statements.

Line 408, "directly linked to the jet coreline", "WCB outflows influence the jet corelines" --  it isn't obvious to me how these methods may accomplish this, unless combined with some dynamical analysis that suggests the causality.

Figure 13, I think the lower panel of this figure would be seriously compromised if viewed in grey scale (you would not be able to distinguish the "coolwarm" type color palette from the grey scale tropopause surface.  You might thus want to think about changes to the presentation here.

Lines 414--415, Per general comment (2b), there is already a vast body of research on these topics, including the few papers I've mentioned here along with many others, covering topics such as relationships of jets and tropopauses to storm tracks, extreme weather events, etc.  Jet regimes of various sorts have been defined based on characteristics of the jet stream, see general point (2b).  While I think the methods in this paper can be a very valuable addition to the existing tools and literature aimed at more fully characterising the jets as dominant features influencing the tropospheric circulation, it is disingenuous to state these solely as future aims.

Lines 419--420, I'm not sure what you mean by a "non-incremental" search, perhaps you can explain this briefly.

**Typos / Grammar / Minor Wording / Etc:**

Line 16, "is" should be "are" ("data" is plural).

Line 17, "time-dependent" should be followed by a comma.

Line 17, "This data is" -> "These data are".

Line 66, I don't think "package" is the best word here; I would just say something like "rotation of the air enclosed between two…"  (in fact if there are diabatic motions, it is not "trapped" in any sense).

Line 84, I would hardly call a paper published in 2001 "recent".

Lines 260 and 262, "is" should be "are".

Line 350, "he jet streaks" should be "the jet streaks".

Line 379, by "attained considerable focus" do you mean it is a topic currently under investigation?  If so, just say that.

Line 421, "Dur" should be "During"

**References not already cited:**
(Apologies for non-uniform format, these are pasted from most convenient sources.)

Danielsen, E. F. (1968), Stratospheric-tropospheric exchange based on radioactivity, ozone and potential vorticity, J. Atmos. Sci., 25, 502–518, doi:10.1175/1520-0469(1968)025 <0502:STEBOR> 2.0.CO;2

Harnik, N., C. I. Garfinkel, and O. Lachmy, 2016: The influence of jet stream regime on extreme weather events. Dynamics and Predictability of Large-Scale, High-Impact Weather and Climate Events, J. Li, Ed., Cambridge University Press, 79–94, https://doi.org/10.1017/CBO9781107775541.007.

Highwood, E. J., B. J. Hoskins, and P. Berrisford, 2000: Properties of the Arctic tropopause. Quart. J. Roy. Meteor. Soc., 126, 1515–1532, doi:10.1002/qj.49712656515.

Holton, J. R., P. H. Haynes, M. E. McIntyre, A. R. Douglass, R. B. Rood, and L. Pfister (1995), Stratosphere-troposphere exchange, Rev. Geophys., 33(4), 403–440, doi:10.1029/95RG02097.

Kunz, A., P. Konopka, R. Müller, and L. L. Pan (2011), Dynamical tropopause based on isentropic potential vorticity gradients, J. Geophys. Res., 116, D01110, doi:10.1029/2010JD014343.

Lee, S., and H.-K. Kim, 2003: The dynamical relationship between subtropical and eddy-driven jets. J. Atmos. Sci., 60, 1490–1503, doi:10.1175/1520-0469(2003)060,1490:TDRBSA.2.0.CO;2.

Lillo, S. P., Cavallo, S. M., Parsons, D. B., & Riedel, C. (2021). The Role of a Tropopause Polar Vortex in the Generation of the January 2019 Extreme Arctic Outbreak, *Journal of the Atmospheric Sciences*, *78*(9), 2801-2821. https://journals.ametsoc.org/view/journals/atsc/78/9/JAS-D-20-0285.1.xml

Lucas, C., B. Timbal, and H. Nguyen, 2014: The expanding tropics: A critical assessment of the observational and modeling studies. Wiley Interdiscip. Rev.: Climate Change, 5, 89–112, https://doi.org/10.1002/wcc.251.

Maher, P., M. E. Kelleher, P. G. Sansom, and J. Methven, 2020: Is the subtropical jet shifting poleward? Climate Dyn., 54, 1741–1759, https://doi.org/10.1007/s00382-019-05084-6.

Manney, G.L., et al., Jet characterization in the upper troposphere/lower stratosphere (UTLS): Applications to climatology and transport studies, Atmos. Chem. Phys., 11, 6115–6137, 2011.

Manney, G.L., M.I. Hegglin, W.H. Daffer, M.J. Schwartz, M.L. Santee, and S. Pawson, Climatology of Upper Tropospheric/Lower Stratospheric (UTLS) Jets and Tropopauses in MERRA, J. Clim., 27, 3248–3271, 2014.

Manney, G.L., and M.I. Hegglin, Seasonal and Regional Variations of Long-Term Changes in Upper Tropospheric Jets from Reanalyses, J. Clim., 31, 423–448, 2018.

Manney, G.L., Z.D. Lawrence, and M.I. Hegglin, Relationships of interannual variability in upper tropospheric jets to ENSO in reanalyses, J. Clim., https://doi.org/10.1175/JCLI-D-20-0947.1, 2021.

Schoeberl, M. R. (2004), Extratropical stratosphere-troposphere mass exchange, J. Geophys. Res., 109, D13303, doi:10.1029/2004JD004525.

Shapiro, M. A. (1980), Turbulent mixing within tropopause folds as a mechanism for the exchange of chemical constituents between the stratosphere and troposphere, J. Atmos. Sci., 37, 994–1004, doi:10.1175/1520-0469(1980)037 < 0994:TMWTFA > 2.0.CO;2.

Spensberger, C., and T. Spengler, 2020: Feature-based jet variability in the upper troposphere. J. Climate, 33, 6849–6871, https://doi.org/10.1175/JCLI-D-19-0715.1.

Stohl, A., et al. (2003), Stratosphere-troposphere exchange: A review, and what we have learned from STACCATO, J. Geophys. Res., 108(D12), 8516, doi:10.1029/2002JD002490.

Winters, A. C., Keyser, D., Bosart, L. F., & Martin, J. E. (2020). Composite Synoptic-Scale Environments Conducive to North American Polar–Subtropical Jet Superposition Events, Monthly Weather Review, 148(5), 1987-2008.
https://journals.ametsoc.org/view/journals/mwre/148/5/mwr-d-19-0353.1.xml

---

## Author Comment (AC2)

**Integration-based Extraction and Visualization of Jet Stream Cores**

**Final Response**

Submission ID: gmd-2021-240

**Geoscientific Model Development**

Dear Reviewers,

We would like to thank you for your constructive and helpful comments. This document presents our response to the questions, and details how we address comments and suggestions in a revised manuscript. In this document, referee questions are written in black, while author replies are written in blue. Throughout this document, several images show a preview of the revised manuscript to illustrate the changes. Revised text passages are highlighted in red.

Sincerely,
The authors.

**Review 1**

*Anonymous Review*

Summary:

This paper proposes a new method to extract jet-stream core lines by using a predictor-corrector approach. Instead of defining the feature as a local extremum point at each grid point, they use an integration-based approach where from precomputed seed point of maximum wind speed the line is traced along the local wind flow and corrected towards the ridge lines to obtain the final core line features.

Their work is based on the local jet core extraction method by Kern et al., but in contrast to Kern's method, their approach does not suffer from cluttered, disconnected features. Instead, they demonstrate that their features remain connected over regions of high wind speed, and align with ridge lines. They are further able to identify merge and split events of the core line features that occur at certain time steps.

Contributions:

- Novel automated method to compute core lines using multiple time steps and a predictor-corrector approach, serves as an extension of Kern et al.'s method.
- Automated identification of split and merge events
- Interactive visualization of these features, along with associated atmospheric processes

In my opinion, this paper shows a scientific contribution to the community, its writing style is good and easy to understand, and it clearly demonstrates the benefit of the proposed method by means of real-case applications. In particular, the authors show, similar to Kern's work, that their approach helps meteorologists to better understand the intercorrelation between jet stream core lines and surrounding / associated atmospheric features. I also want to highlight the short but good explanation of potential vorticity, warm-conveyor belts, tropopause, and the core line feature itself. There are only minor suggestions or questions from my side, but I can recommend accepting this paper with some minor corrections.

Critics:

- Figure 5 and Figure 6 should also contain the color tables, or it should be explained what the color means. The general captions of the figures are good, but some of the color tables are hard to read (Figure 8). I would recommend using larger text fonts or annotate the tables with latex.
**We added the annotations for Figures 5 and 6. Throughout the document, all color maps and their annotations are now placed with LaTeX to keep the font sizes consistent with the text. In the following, the placement of color maps is shown for Figures 5, 6 and 8.**

[Figure]

jet speed (ms$^{-1}$)

110

**(a)** $n_{stepsBelowThresh} = 0$      **(b)** $n_{stepsBelowThresh} = 100$      **(c)** $n_{stepsBelowThresh} = 1500$

**Figure 5.** Increasing the number of prediction steps a jet core might remain below the wind magnitude threshold, results in longer connected jet corelines. Here, for 11.09.2016 15:00.

[Figure]

jet speed (ms$^{-1}$)      wind mag. (ms$^{-1}$)

110      10    110

**(a)** $n_{corrSteps} = 1$      **(b)** $n_{corrSteps} = 5$      **(c)** $n_{corrSteps} = 12$

**Figure 6.** Comparison of jet corelines for varying number of correction iterations. The higher the number, the closer the line follows a ridge line as proposed by Kern et al. (2017), which might exhibit higher curvature. Lowering the number of corrector iterations smoothes the line. A cross section of the wind magnitude field shows how well the extracted corelines pass through sectional extrema. Here, for 11.09.2016 15:00.

[Figure]

jet speed (ms$^{-1}$)      wind mag. (ms$^{-1}$)

110      40    109

**(a)** *Jet stream with wind magnitude volume rendering*      **(b)** *Warm conveyor particles with pathlines*

**Figure 8.** Jet stream corelines are rendered as tubes, with jet speed being mapped to color and radius. Left: with volume rendering of wind speed, Right: warm conveyor belt particles are shown as particles with pathlines attached to convey motion.

- Typo in line 350: "jhe" --> "the"
**Fixed.**
- Table 1: What do Var1 and Var2 mean?
**We listed extraction timings for the default parameters and two alternative parameter**
**settings. We now rephrased "var" to "variation" and explain the meaning in the caption**
**of Table 1: *"[...], here listed for the default parameters and two variations from the default***
***parameters."***
- In Figure 4, the authors compare the parallel vectors approach with their proposed method,
however, earlier in the text, they emphasize that their work is based on the method from Kern et
al. Are the results similar to the parallel vectors approach? Or can it be re-formularized using the
parallel vectors operator? Maybe the authors could also show the effect of smoothing and how
much the features actually diverge from the target result.
**Regarding the parallel vectors reformulation of Kern:**
**Eq. (3) is an equivalent reformulation of the Kern feature definition from Eq. (2) into the**
**parallel vectors notation. Two vectors are parallel, when their cross product produces**
**the zero vector. Expanding the cross-product yields the two equations from Eq. (2) and**
**the third condition 0=0, which is always fulfilled.**

$$\begin{pmatrix} \partial s/\partial \mathbf{n} \\ \partial s/\partial \mathbf{z} \\ 0 \end{pmatrix} \parallel \begin{pmatrix} 0 \\ 0 \\ 1 \end{pmatrix} \Leftrightarrow \begin{pmatrix} \partial s/\partial \mathbf{n} \\ \partial s/\partial \mathbf{z} \\ 0 \end{pmatrix} \times \begin{pmatrix} 0 \\ 0 \\ 1 \end{pmatrix} = \begin{pmatrix} 0 \\ 0 \\ 0 \end{pmatrix} \Leftrightarrow \begin{pmatrix} \partial s/\partial \mathbf{z} \\ \partial s/\partial \mathbf{n} \\ 0 \end{pmatrix} = \begin{pmatrix} 0 \\ 0 \\ 0 \end{pmatrix} \tag{3}$$

**We expanded the equation and now explain this after Eq. (3): *"The symbol $\|\|$ denotes***
***the parallel vectors operator (Peikert and Roth, 1999), which receives two vector fields as***
***input and produces the set of points at which the two given vector fields are parallel. The***
***two vectors are parallel if their cross product vanishes to zero. Applying the cross***
***product results in three equations: the two equations from Eq. (2) and $0=0$."***
**Regarding the differences between Kern and the predictor-corrector approach:**
**For a high number of corrector steps, our approach converges to the ridge line of Kern et**
**al. (2017), as both methods aim for the same feature definition (wind magnitude extrema).**
**By controlling the number of correction steps of the parallel vectors extractor, the lines**
**can be regularized to follow the prediction direction, which results in smoother lines.**
**Regarding the effect of smoothing:**
**In our work, the amount of smoothing is controlled by the number of corrector steps. The**
**more corrector steps are applied, the more the jet is aligned with the ridge line in the**
**wind magnitude field, which is the feature that Kern et al. extracted. Figure 6**
**demonstrates the effect of varying the number of corrector steps. We added more**
**explanations to the caption to make clear that this parameter controls the smoothness:**
***"The higher the number (of correction iterations), the closer the line follows a ridge line***
***as proposed by Kern et al. (2017), which might exhibit higher curvature. Lowering the***
***number of corrector iterations smoothes the line."***
- Extremum lines in general do not have to be aligned with the flow. However, the authors
actually want the features to follow the local streamlines if I understood it correctly. What is the
intention here? Is it due to numerical instability and grid resolution that integrating the lines
along the flow leads to more accurate results?

**Yes, the low vertical grid resolution leads to unnatural bending of the ridge lines in the**
**vertical direction, see Fig. 6c. The alignment with the wind direction serves as a**
**regularization. We now explain the reasoning in the introduction section to better**
**motivate the approach:** *"The latter (flow alígnment) serves as regularization to prevent*
*unnaturally bent ridge lines caused by a low vertical resolution."*
- Why did the authors choose to perform a regridding of the hybrid model level data? One could
also extract the feature directly from model levels, however, gradients and interpolation must be
done differently. Is it just because of simplicity or due to the focus on the tropopause and the
upper pressure levels? For feature extraction near the surface, model levels might be more
suitable than interpolated pressure levels.
**The regridding was done for computational convenience. The regridding led to 10x more**
**grid points, i.e., for every hybrid model level, we placed 10 regular grid points in the**
**vertical direction. During development, we went up to 30 regular grid points per hybrid**
**model level to be sure that no differences occur when increasing the grid resolution**
**further. Regridding consumes additional memory, which can be avoided by working**
**directly on the hybrid model level data. As mentioned by the reviewer, this requires**
**adjustments in the calculation of partial derivatives and interpolation. We added this**
**discussion to the data section.**
- Figure 11: The core lines and the surface can hardly be seen. Would it be possible to use a
more detailed view and a different viewing angle? Especially the top image of 11.a) does not
clearly depict the features.
**In addition to the top view, we now also provide a side view for Fig 11a, as shown below.**
**Further, we added zoom-ins that display jets in the southern hemisphere. Additional**
**camera angles for 11b can be seen in the accompanying video. An additional three-**
**dimensional view in the equirectangular projection follows later in Figure 13, where the**
**corelines and their relative positioning to the tropopause can be seen better.**

[Figure]

**(a)** *Equirectangular projection at 01.09.2016 00:00. (top view)*   **(b)** *Equirectangular projection at 01.09.2016 00:00. (side view)*

[Figure]

**(c)** *Polar stereographic projection at 03.09.2016 18:00.*

**Figure 11.** Interaction of Jet stream and tropopause. The jet stream is located where the tropopause is steep. Jet speed is mapped to tube color and tube radius.

General questions:

- Is the predictor-corrector approach more stable for coarser grids than the other local methods? And what about more fine-scale grids?
**The predictor-corrector approach allows the ridge lines to be regularized. When the ridge line exhibits high vertical curvature, then regularizing with a smooth vector field helps to produce smoother lines. The high vertical curvature of the ridge line is a product of the low vertical resolution of the hybrid model levels. With an increased model resolution, such regularization will hopefully not be necessary anymore in the future. Apart from this, local feature extractors such as parallel vectors often experience fragmentation independent of the discretization of the domain, resulting in spurious lines that have to**

**be reconnected in a post-process. Predictor-corrector approaches are in the class of**
**integration-based methods, which generally avoid this numerical issue. We added this**
**discussion to the conclusion.**
- I would also suggest improving the conclusion and clearly demonstrate the benefit of the
proposed method. What is the improvement over existing methods? Kern et al also
demonstrated its benefit for operational forecasting. Is your approach and visualization tool able
to help forecasters in operational service?
**The benefits of the predictor-corrector approach are the ability to regularize the line**
**geometry and the inherent long connectivity of the extracted feature lines. With our**
**previous answer, these benefits are now stated more clearly in the conclusions. While**
**the extraction algorithm could be applied to data arising in an operational context, more**
**work is necessary for a successful integration in operational routines, including an**
**increased temporal stability, heuristics for automatic parameter selection, and a**
**requirement analysis with operational forecasters to integrate potential additional**
**constraints into the feature definition and extraction. We appended this interesting**
**avenue for future work in the conclusions.**

**Review 2**

*Gloria Manney*

**General Comments:**
This paper presents a new method and software for tracking and visualizing jet stream cores. This is a potentially very useful new method with some important advantages and should be a valuable addition to existing tools for jet characterization and analysis. As such, it should be appropriate for publication in GMD and I would expect there to be much interest in it among atmospheric scientists who focus on studies of the jet stream (including myself!). However, I feel there are some important changes to the presentation needed to (1) better reflect previous work on jet stream characterization and the phenomena (WCBs, tropopause structure) the jet streams are related to here, and (2) to make the paper more accessible to an audience of atmospheric scientists for whom this method / software may be very useful but who may not in general be computer scientists or mathematicians. (I believe I'm a reasonable example of this class of atmospheric scientist, so that if I don't understand some things it is not unlikely that many other interested readers will be in the same position.) These changes are summarized here (with some further specific examples given in the "specific/minor comments" below):

- (1) A few terms and some notation are used throughout this paper that are not (clearly) defined or are things many in your audience may not be familiar with, and should be defined and/or expressed in plainer language:
  - (a) Voxel -- should just be defined the first time it is used as it will be unfamiliar to many readers (as I understand it from looking up the definition, it is nothing more than the 3-D analog of a pixel).
    **In Section 3.3, we now first introduce the terminology (the domain is discretized onto a grid composed of cells). We now avoid the use of the word voxel.**
  - (b) Heuristic (heuristics, heuristically) -- in general (and in many fields) this term is often (perhaps over) used and frequently not clearly defined (thus sometimes mis-used). Indeed, dictionary definitions are many and varied. My impression is that the way you use it here is something akin to "pertaining to a trial-and-error method of problem solving used when an algorithmic approach is impractical", or to simply say that the process in question requires human intervention (e.g., the necessity to make choices based on things the human eye does very well but we tend to have trouble telling computers how to do). A more specific statement (and perhaps examples from some of the previous studies you cite) of what you mean by "heuristic" would be very helpful in motivating the development and advantages of your method.
    **We avoided the word heuristics and now explained the local method in more detail to point out the disadvantages that the proposed approach avoids. We now explain that local line extraction methods solve for lines in three steps. First, intersection points with the cell boundaries are computed numerically per cell. Second, the intersection points are connected to form line segments within the cells, which may fail if intersection points were missed or are duplicated due to numerical reasons. Third, the line segments are connected to continuous lines when the end points of two segments are close enough to each other (within a threshold) and when the tangent directions at the end points align (up to a certain threshold). The result of this last operation is order-dependent,**

| 200 | **depends on the numerical accuracy of the first step, and is dependent on** |
| 201 | **thresholds.** |
| 202 | o (c) Manifolds -- I question the need to use this term (which readers unschooled in |
| 203 | topology may not recognize or may immediately assume is expressing some |
| 204 | complicated concept) when the fundamental information conveyed in this context |
| 205 | by "instantaneous 1-manifolds" is that it is a line/curve (1-dimensional) at a |
| 206 | particular time, and by "time-dependent 2-manifold" that it is a time-varying |
| 207 | surface (2-dimensional). |
| 208 | **We agree, the usage of the topological terms is not necessary, since the** |
| 209 | **number of independent variables can be inferred from the terms "curve"** |
| 210 | **and "surface". We rephrased "1-manifold" to "curve" and "2-manifold" to** |
| 211 | **"surface".** |
| 212 | o (d) Several definitions (tropopause, WCBs, and "filtered" WCBs) used are |
| 213 | expressed in set-builder notation (which many readers may not be familiar with); |
| 214 | in general, you explain these (though not always completely) in words |
| 215 | beforehand, but it isn't always obvious that that is what you are doing. I'd suggest |
| 216 | that you make this relationship explicit, by saying something like (e.g., for the |
| 217 | tropopause case): "...largest connected surface of 2 PVU (-2 PVU) in the |
| 218 | northern hemisphere (southern hemisphere) at pressures below 740 hPa, that is: |
| 219 | " (in fact, since this is a very common way to define the |
| 220 | tropopause, any atmospheric scientist interested in your work will immediately |
| 221 | understand this without the equation, so it is not obvious that you need the |
| 222 | equation at all -- however, I have no problem with including it as long as it is also |
| 223 | explained in plain words so that the reader will understand regardless of whether |
| 224 | they are familiar with the notation. |
| 225 | **We rephrased the paragraph of the tropopause definition, such that we first** |
| 226 | **explain the definition by words, and then the formal definition is given,** |
| 227 | **stating that this is the same definition but in formal language. The** |
| 228 | **definitions of WCB trajectories and WCB-tropopause intersections are** |
| 229 | **written in the same way.** |
| 230 | • (2) The citation of previous literature is lacking, especially with regard to upper |
| 231 | tropospheric jet streams and characterisation thereof. While I understand that the focus |
| 232 | here is on the software tools developed, these are being presented specifically in |
| 233 | relation to jets in the Earth's atmosphere and thus the primary audience is atmospheric |
| 234 | scientists -- hence it is important to accurately relate this work to previous work in the |
| 235 | field and to the reasons why the method may facilitate future work. In particular: |
| 236 | o (a) Section 2.2: There are much better references than Dameris (2015) for what |
| 237 | the tropopause is and why it is important; in addition, Dameris (2015) is not |
| 238 | readily publicly available. I would start with the reviews by Holton et al (1995, Rev |
| 239 | Geophys) and Stohl et al (2003, JGR). In addition to Skerlak et al (2015), I would |
| 240 | add a couple of classic papers for tropopause structure such as (cited by Skerlak |
| 241 | et al) Danielsen (1968) and Shapiro (1980) -- or at least add "and references |
| 242 | therein". Highwood et al (2000, QJRMS), Schoeberl (2004, JGR), and Kunz et al |
| 243 | (2011, JGR) are good references for the range of PV values that have been used |
| 244 | for dynamical tropopause identification and for what regions / purposes different |
| 245 | values are appropriate. |
| 246 | **We included the suggested references in Section 2.2 to guide the reader for** |
| 247 | **more details to the related work.** |
| 248 | o (b) Section 2.3: This is a very incomplete and biased discussion of upper |
| 249 | tropospheric jets and previous work characterizing them. |

▪ (i) Ahrens and Henson (2018) is not readily publicly available, and there
are numerous choices for classic work describing the jet streams and
their importance to the atmospheric circulation. Koch et al (2006) and
Schiemann et al (2009) (already cited in this preprint) both give concise
historical introductions. Harnik et al. (2016) provide a nice brief review in
relation to jet regimes and extreme weather events.. Manney et al (2014,
J Clim) and Manney and Hegglin (2018, J Clim) have in their introductions
comprehensive discussions of the literature in the context of the
importance of and variations in jet streams, which provides many of those
classic references.
**We included the suggested references in Section 2.3 to motivate the**
**analysis of jet streams.**
▪ (ii) Several methods for identifying and characterizing jet streams that
seem very relevant to this work are not mentioned, including the method
introduced by Manney et al (2011, ACP) and used in Manney et al (2014,
2021, J Clim) and Manney and Hegglin (2018) (with more physically-
based distinctions of the subtropical and polar jets in the latter two of
those papers); the method used by Winters et al (2020, MWR, and
several references therein); and that of Maher et al (2020, Clim Dyn).
Should also cite Spensberger & Spengler (2020, J Clim) in addition to
Spensberger et al (2017) and note that their method does the
characterization on the dynamical tropopause.
**We included the suggested references in Section 2.3 when**
**introducing jet extraction methods.**
▪ (iii) There are many jet characterization methods / studies in which the
"assumption that the flow is oriented eastwards" is not made (and others
where it is only used after the fact), including Manney et al (2011 ACP,)
(method also used by Manney et al (2014) and refined for Manney and
Hegglin (2018) and Manney et al (2021).) In addition several methods
(including that of Manney et al, above references) do characterize the jet
position/extent in the vertical as opposed at a level or in a layer.
**We added the references and briefly summarized the methods,**
**giving credit to the vertical consideration.**
▪ (iv) The statement that the jets "can be further classified into different
types based on their location" is vastly oversimplified, and, given the
usage of these terms later in the paper, the physical distinction between
subtropical and polar jets should be discussed accurately here, as well as
the fact that there is indeed a spectrum of jets that may have
characteristics that are a hybrid between the two (see Lee & Kim, 2003,
JAS; Manney et al, 2014, 2021; Winters et al, 2020; and references
therein). Peña-Ortiz et al (2013), in fact, noted that attempting to
distinguish polar and subtropical jets by latitude was commonly
unsuccessful; Manney et al (2011, 2014) noted that using a simple
latitude criterion was only useful for very broad climatological studies, and
Manney and Hegglin (2018) introduced a more physically-based method
of distinguishing subtropical and polar jets based on tropopause height
changes across the jet region. Winters et al (2020, and references
therein) distinguish subtropical and polar jets by identifying them in
different isentropic layers, and show clear instances of them merging to
form a jet with hybrid characteristics.

**We included the summary of classification methods as suggested.**
**In the remainder of the manuscript we no longer distinguish jet**
**types and concentrate on the extraction of their coreline geometry**
**instead.**

○  (c) I am not as familiar with the literature on WCBs, but the discussion strikes me
as often making general statements without giving citations (some instances
noted below in the specific comments).
**We added references to support the statements. The individual instances**
**are described further below in the specific comments.**
• (3) General Questions (those with ** at the beginning are more further information /
general interest questions, rather than necessary changes to this manuscript):
○  (a) Rationale for choices of thresholds/definitions, and discussion of sensitivity to
those thresholds/definitions (there is a statement on line 279 that "thresholds
used in the definitions have been chosen based on common practices in
atmospheric science", but you need to give references and briefly note the
physical basis for that "common practice", including:
**Instead of "common practice", we now write that the parameters are**
**chosen empirically. We mention other choices as suggested below.**
▪  (i) 40 m/s for the minimum windspeed for the seed points. (Some
information on the sensitivity of the performance of the software to this is
given, but no rationale is given for the default choice, nor is the sensitivity
of the physical results to this discussed.)
**For the proposed jet coreline extraction method, this threshold is an**
**algorithmic choice, and results for different options have been**
**shown. A discussion of the physical consequence of different**
**threshold choices goes beyond the scope of the paper, as this**
**cannot be done without an analysis of the dynamic processes. We**
**added to the manuscript that 40m/s was chosen empirically.**
▪  (ii) 190 to 350 hPa for the domain for jet extraction. 190 hPa is not low
enough to exclude the stratospheric "subvortex" jet, which commonly
extends down to between 150 and 250 hPa (eg, 340K), especially in the
SH late winter and spring (e.g., Manney et al., 2014). Manney et al (2014,
their Fig. 6) fairly commonly identified jet cores of over 40m/s as low as
about 5 km, near or at their high pressure search boundary of 400 hPa,
as well as jet cores (distinct from the stratospheric subvortex jet, which
they characterized separately) near 13--14 km (typically for subtropical
jets at latitudes equatorward of about 30 degrees), very near or at their
low pressure search boundary of 100hPa; as they noted (and identified),
the stratospheric subvortex jet often overlaps considerably in altitude with
the upper tropospheric jets.
**Section 3.1 reports the data and the search bounds (190 to 350 hPa)**
**that we used when developing the algorithms on our two month time**
**window. We now note that the search space needs to be increased,**
**depending on the analysis task and the considered spatial and**
**temporal domain, referring to the work of Manney et al. (2014) for jet**
**extractions below and above our considered pressure range.**
▪  (iii) 2 PVU for the dynamical tropopause (many other values are used,
and higher values of 3 to 5 PVU have often been recommended for mid to
high latitude features, eg, Highwood et al, 2000, QJRMS; Schoeberl,
2004, JGR; Kunz et al, 2011, JGR), and 740 hPa for the maximum
pressure (e.g., intense sub-synoptic scale events can be associated with

| 351 | the tropopause dropping well above this pressure, eg, Lillo et al, 2021, |
| 352 | JAS, and references therein). Also, if the domain studied extends into the |
| 353 | tropics, how is the tropopause computed there since PV goes to zero (the |
| 354 | most common procedure is to use an isentropic surface, commonly 380K, |
| 355 | wherever the magnitude of the PV is less than the threshold above this |
| 356 | isentropic level, eg, Schoeberl, 2004; Manney et al, 2011; and references |
| 357 | therein)? |

351. the tropopause dropping well above this pressure, eg, Lillo et al, 2021,
352. JAS, and references therein). Also, if the domain studied extends into the
353. tropics, how is the tropopause computed there since PV goes to zero (the
354. most common procedure is to use an isentropic surface, commonly 380K,
355. wherever the magnitude of the PV is less than the threshold above this
356. isentropic level, eg, Schoeberl, 2004; Manney et al, 2011; and references
357. therein)?

358. **We now mention the different choices for the PV isosurfaces and the**
359. **pressure threshold, when introducing the formal definition for the**
360. **tropopause. We extract isosurfaces for +2 and -2 pvu separately and**
361. **display both surfaces together. Along the equator, we did not handle**
362. **the sign flip, since we did not concentrate on tropical regions. We**
363. **now refer to the work of Schoeberl and Manney in this context.**

364. ▪ (iv) Thresholds for proximity of WCBs to jet coreline.
365. **We now mention that those thresholds are chosen empirically.**
366. **Those parameters are not part of the jet coreline computation, but**
367. **are used to define spatial proximity of WCBs and jet corelines.**

368. ○ (b) I find the discussion overall somewhat unclear in the usage of "steps" -- there
369. are spatial steps (eg, the grid spacing used for the prediction step), time steps,
370. and procedural steps (eg, prediction and correction steps) and it is not always
371. clear from the context which is being discussed.
372. **We carefully checked the manuscript for all occurrences of "step" and**
373. **clarified whether these are "time steps", "integration steps", "correction**
374. **steps" or "prediction steps".**

375. **.**

376. ○ (c) The representation of the corelines (which, as I understand it, are simply that,
377. that is lines approximately connecting the core locations) as tubes, with wider
378. tubes for higher windspeeds has the potential to confuse the reader into believing
379. they show the jet region (analogous to the "regions" discussed in Koch et al,
380. 2006 and Manney et al, 2011). While there will be some information since
381. regions with higher windspeeds will have windspeeds above the threshold(s)
382. over a larger area, there is by no means a direct correspondence since the wind
383. gradients are not uniform or symmetrical around the core. The text needs to be
384. very clear about this point, so as to not mislead the reader into thinking they are
385. seeing the physical region where a jet is defined.
386. **For all figures in which the velocity magnitude was mapped to the tube**
387. **radius, we now explain this encoding in the caption.**

388. ○ (d) While the terms subtropical and polar jets are tossed around in the paper
389. there is apparently no attempt to distinguish these in a physically meaningful
390. way. Thus statements suggesting that a jet coreline represents a subtropical or
391. polar jet should not be made. **Also, it would be interesting to know if there are
392. plans to add such a distinction to the method.
393. **Now we only mention in Section 2.3 (Jet Streams) that different jet stream**
394. **types exist. In the remainder of the paper, we no longer distinguish their**
395. **types. Identifying the type of a jet given the jet geometry would certainly be**
396. **interesting.**

397. ○ (e) Use of pressure rather than potential temperature for the vertical coordinate:
398. Why was the pressure coordinate chosen for the jet extraction? Given that an
399. isentropic coordinate would be more "flow-following" on short (days to a week or
400. two) timescales, would one expect substantial differences if the procedure were implemented in an isentropic coordinate? **Would it be feasible to implement it in
isentropic coordinates?
**PV can be defined in both coordinate systems and subcommunities have**
**different preferences. Conceptually, the predictor-corrector based**
**extraction is possible in both coordinate systems, since both the prediction**
**and the correction follow ODEs that can be equivalently expressed in**
**different coordinates. We mention this now in Section 3.1.**
o  (f) **I would be interested in some more discussion (perhaps largely in an
appendix or in the supplemental material) on the performance. The description
given is all per time step (and it is not entirely clear what the time step being
referred to is). Your study period is two months. What is the total time to process
that period? The description of the procedure sounds storage-intensive -- what is
the total storage needed for output for your study period? What do the
performance results imply about the feasibility of using this procedure for
climatological studies? From all of this, can you say something about the system
requirements (CPUs/speed, memory, cache, storage) for running this
effectively?
**We now clarify in Section 3.4 that we worked with hourly simulation data.**
**The computation time is listed per simulation time step and in total for the**
**whole two months of simulation data. The used processor is listed, as well.**
**The code uses basic OpenMP parallelization, but is not optimized for cache**
**efficiency, memory usage and low storage requirements.**
o  (g) **I would also be interested (again, in an appendix or supplementary material)
in more specifics about the algorithms used for various steps. I think such
information could be very helpful to the reader who might want to implement
something similar to parts of this but is not conversant with C++.
**The main ingredients for a reimplementation of the predictor-corrector**
**approach are:**
1. **Interpolation of variables from a discrete grid (we used trilinear**
**interpolation).**
2. **identification of extremal points for seeding (find grid points around**
**which all adjacent grid points have a lower wind speed)**
3. **Numerical integration of an ODE (we used a fourth-order Runge-**
**Kutta integration)**
**These details are now described in Section 3.2.**
**Specific / Minor Comments (in order of appearance):**
Line 22, please provide (a) more accessible and foundational reference(s) per general comment
(2).
**We added previously suggested references that introduce jets and emphasize their**
**importance.**
Line 28--29, why is the tropopause expected to show highly 3-d structures around split and
merge events? Please give references for this.
**We rephrased this to make clear that this is not necessarily expected, but instead a**
**hypothesis that we want to investigate by extracting and visualizing the jets and the**
**tropopause in 3D. This sentence serves as motivation to look at these structures in 3D.**
Line 76--80, would be good to note somewhere in here that the tropopause altitude is generally
highest in the tropics, lowest near the poles, and drops sharply across the subtropical jet. It not uncommonly extends below 6km in folds or other tropopause depressions (see general
comment (2a)).
**We added this general remark as suggested and included the reference to Lillo et al.**
Line 105, this is presumably a right-handed coordinate system, and v is defined as positive if
Northward?
**Indeed, we added that u is oriented eastward and v is northward. Left- or right-**
**handedness depends on the direction of the vertical axis k = (0,0,1). Both orientations are**
**possible, since the parallel vectors condition in Eq. (3) results in root-finding problems,**
**which have the same solution regardless of whether the axis is multiplied by -1.**
Lines 108--112 (through eq. 3), this sentence / equation aren't very clear. I'm guessing that the
text is supposed to be a description of the following equation, but I don't know what the || means
in this context (where it looks like an operator or something stating a relationship) nor whether
the equation is supposed to represent the rephrasing of the problem or something related to the
solver. Please re-word.
**We added more detail to explain how Eq. (3) is a reformulation of Eq. (2). The parallel**
**vectors operator "||" is indeed an operator that receives two vector fields as input and**
**produces the set of all points at which the two given vector fields are parallel. Two**
**vectors are parallel if the cross-product is zero. The cross product has three vector**
**components. The first two components are the expressions of Eq. (2) and the third**
**component gives 0. There are a number of standard algorithms to find the roots of those**
**cross-product components.**
Figure 1 caption, please clarify that the date/time at the end of the caption is that shown in the
Figure.
**Yes, the time in the caption is correct. By closer inspection we noticed that the image**
**was vertically flipped, which is now corrected. In this image, weak jets over Asia are**
**shown.**
Lines 121-122, some general reference(s) should be given for cyclones.
**We included the following references on cyclones to refer the reader to a more elaborate**
**introduction:**
**(1) Wernli, H. and Schwierz, C.: Surface Cyclones in the ERA-40 Dataset (1958–2001). Part I:**
**Novel Identification Method and GlobalClimatology, Journal of the Atmospheric Sciences, 63,**
**2486 – 2507, https://doi.org/10.1175/JAS3766.1, 2006.**
**(2) Schultz, D. M., Bosart, L. F., Colle, B. A., Davies, H. C., Dearden, C., Keyser, D.,**
**Martius, O., Roebber, P. J., Steenburgh, W. J., Volkert, H.,and Winters, A. C.: Extratropical**
**Cyclones: A Century of Research on Meteorology's Centerpiece, Meteorological**
**Monographs, 59, 16.1 –16.56, https://doi.org/10.1175/AMSMONOGRAPHS-D-18-0015.1,**
**2019.**
Line 192, please explain what is meant by easing "the balancing between prediction and
corrections steps" and why normalization accomplishes this.
**The rate of how fast a numerical integration proceeds through space depends on the**
**magnitude of the velocity and the chosen integration step size (which is constant). By**
**normalizing the velocity vector, the amount of spatial movement only depends on the**
**number of prediction steps $n\_predStep$ and the number of correction steps**
**$n\_corrSteps$. That is, it no longer depends on the wind speed. We added more**
**explanation for this after Eq. (5).**

Line 214, and Figure 2 caption. If the weak endings are removed, why are there still green segments in Fig. 2(d)?

**The green segments in Fig 2d are regions on the jet at which the wind velocity magnitude threshold is temporarily not reached. When tracing a jet, we terminate the jet only when this happens for more than $n\_stepsBelowThresh$ subsequent integration steps. With this, the jets are allowed to temporarily fall below the threshold and remain connected for a longer time. We added an explanation to the figure caption.**

Line 292, what is the method for integrating dx(t)/dt for the trajectories?

**We use a fourth-order Runge-Kutta integrator. We added this to Eq. (7).**

Lines 316-317 & 331, I don't know what you mean by "transfer functions", please explain or reword (since it sounds like you are just saying both the radius and color are dependent on magnitude, you could simply say that). It would be helpful to have some sort of a key for the radius on the plots; if the radius relationship is also linear (as the color one appears to be) you might make the color bar a wedge rather than a rectangle. If the radius change is not linear with windspeed, you need to say that. Related to this, and Figs. 10, 11, and 12, it needs to be explicitly stated that the radius does not show the region wherein a jet is defined, per general comment (3c).

**We clarified in Section 4.3 (Visual Mapping) that color and tube radius are dependent on the velocity magnitude. The term "transfer function" is standard terminology in scientific visualization and refers to the mapping of a quantitative attribute to a visual channel, for example a color, a transparency or a radius. As suggested, we made the color bar to a wedge rather than a rectangle in all figures, in which the magnitude was mapped to the tube radius. In addition, we explicitly mention in the respective captions that the radius is determined by the magnitude. Here is an example from Figure 8(a):**

[Figure]

Line 318, how did you determine that it was a stratospheric jet?

**The jet was positioned above the tropopause. As mentioned earlier, we have now removed all classifications of jets, as this is a separate topic.**

Line 326, why not use a north polar orthographic or a north polar Lambert equal area projection? These emphasize the mid to high latitude regions more than the stereographic.

**We now mentioned in Section 4.3 (Viewing Projections) that other projections are imaginable as well.**

Figures 2, 8, 9 10, and 12, the color bars are too small. Also, the choice of a diverging color
palette for the windspeed in these and other figures seems a poor one, since it is a positive
definite quantity for which it does not appear that there is a reason to emphasize a transition at
one particular value -- a perceptually uniform sequential palette would be preferable.
**We unified the size and placement of all color bars throughout the paper. We now use a**
**sequential color map for the wind speed whenever we show vertical slices. We would**
**prefer to keep using a diverging color map for the coloring of the jet tubes, since we want**
**to set the reader's attention to the weak jet parts as well, since those are the structures**
**that are affected by temporal incoherence the most. For the purpose of demonstrating**
**strengths and weaknesses of the algorithm, we think that the weak jets structures should**
**not be hidden, and would therefore prefer to keep the diverging color map for jets.**
Lines 340-343, per general comment (3d), you have not done anything to identify polar vs
subtropical jets, which have different primary driving mechanisms and thus different
characteristics (e.g., Lee & Kim, 2003; Manney et al.,2014; Winters & Martin, 2020; and
references therein). There are not "generally" two jets, in fact the patterns of jets and how many
there are (with one to three being most common, but more possible at a given time/longitude)
vary strongly with region and season (e.g., Manney et al., 2014, 2021, especially see Fig. 1 in
the latter). If you are going to use the terms, you need to provide some justification for referring
to a particular jet as polar or subtropical since there are important physical distinctions between
the two (and of course, some jets may have hybrid characteristics between the two).
**Agreed, we removed the statements in lines 340-343. We no longer distinguish between**
**jet types, as this is not the focus of our work.**
Line 355--363, it would also be good to cite Winters & Martin (2020) and Maher et al (2020)
(and references therein) here, since the methods they use (unlike Koch's) rely on those strong
PV gradients. There is a large body of work (much of it cited in these recent papers; also see
Manney et al, 2014, and references therein) showing that extratropical westerly jet cores lie
near/at the dynamical tropopause in the region where there are rapid altitude decreases in the
tropopause altitude with increasing latitude, thus the "expectation" of it lying on the flanks of
valleys in the tropopause, and of tropopause folds "wrapping" along the flank of a jet are
well-known results, as is the complex 3D structure of the tropopause. Per general comment
(2b), there are many papers (including those cited previously in this review) that characterize
the jet structure in both the horizontal and vertical, so the largely 2D views described in lines
362--363 are by no means "typical" and have not been for on the order of the last decade.
**We added the suggested references to give credit to the observed link between 3D folds**
**and jet stream paths. Further, we removed the 2D statement from lines 362-363.**
Lines 366--367, can you say anything about how the visualization (which, though informative
and interesting, is qualitative) will help shed light on mechanisms.
**Visualizations are meant to convey visual impressions of data, enabling researchers to**
**phrase further hypotheses and research questions. Those questions would then be**
**investigated by means of a dynamical analysis of the physical processes. The results of**
**that could then be visualized again to communicate the findings. In other words,**
**visualization is not meant to replace a dynamical analysis, but is a tool aiding in the**
**process. We rephrased the corresponding paragraph accordingly.**
Line 371, why 270 hPa? Also, why on an isobaric rather than an isentropic surface?

**We extracted the jet corelines in isobaric coordinates and hence it was straight-forward**
**to compare the geometry with a horizontal isobaric slice. Conceptually, it is possible to**
**switch to an isentropic coordinate system and show the isentropic surface instead.**
Lines 372 & 378, is this really entirely an effect of the WCB on the jet? That is, is there no effect
of changes in the jet on the WCB? How do you know which is causing which to change?
**We rephrased the sentence to state that the displacement of the jet occurs in the**
**presence of the WCB. A causal connection is not implied, as this would require further**
**investigation of the atmospheric dynamics.**
Lines 373--375, some references for these effects are needed.
**We included the following references to support the discussion on the relationship**
**between WCBs and jets.**
**(1) Oertel, A., Boettcher, M., Joos, H., Sprenger, M., and Wernli, H.: Potential vorticity**
**structure of embedded convection in a warm conveyor belt and its relevance for large-**
**scale dynamics, Weather and Climate Dynamics, 1, 127–153, https://doi.org/10.5194/wcd-**
**1-127-2020, 2020.**
**(2) Joos, H. and Forbes, R. M.: Impact of different IFS microphysics on a warm conveyor**
**belt and the downstream flow evolution, QuarterlyJournal of the Royal Meteorological**
**Society, 142, 2727–2739, https://doi.org/https://doi.org/10.1002/qj.2863, 2016.**
**(3) Blanchard, N., Pantillon, F., Chaboureau, J.-P., and Delanoë, J.: Mid-level convection**
**in a warm conveyor belt accelerates the jet stream,Weather and Climate Dynamics, 2, 37–**
**53, https://doi.org/10.5194/wcd-2-37-2021, 2021.**
Lines 385--386, it would be appropriate to cite Manney et al (2014, 2021), Homeyer & Bowman
(2013, JAS), Winters & Martin (2020), Spensberger & Spengler (2020), and references therein
here, per general comment (2b).
**We included the suggested references.**
Lines 404--407, please provide some references for these statements.
**In the conclusions, we add the following references that discuss the WCB outflows:**
**(1) Grams, C. M., Magnusson, L., and Madonna, E.: An atmospheric dynamics**
**perspective on the amplification and propagation of forecast error in numerical weather**
**prediction models: A case study, Quarterly Journal of the Royal Meteorological Society,**
**144, 2577–2591,https://doi.org/https://doi.org/10.1002/qj.3353, 2018.**
**(2) Spreitzer, E. J.: Diabatic processes in mid-latitude weather systems - a study with the**
**ECMWF model, Ph.D. thesis, ETH Zurich, Zurich,https://doi.org/10.3929/ethz-b-000438728,**
**2020.**
**(3) Saffin, L., Methven, J., Bland, J., Harvey, B., and Sanchez, C.: Circulation conservation**
**in the outflow of warm conveyor belts and consequences for Rossby wave evolution,**
**Quarterly Journal of the Royal Meteorological Society, p. in print,**
**https://doi.org/https://doi.org/10.1002/qj.4143, 2021.**
**(4) Grams, C. M., Wernli, H., Böttcher, M.,ˇCampa, J., Corsmeier, U., Jones, S. C., Keller, J.**
**H., Lenz, C.-J., and Wiegand, L.: The key role of diabatic processes in modifying the**
**upper-tropospheric wave guide: a North Atlantic case-study, Quarterly Journal of the**
**Royal Meteorological Society, 137, 2174–2193,**
**https://doi.org/https://doi.org/10.1002/qj.891, 2011.**

Line 408, "directly linked to the jet coreline", "WCB outflows influence the jet corelines" -- it isn't
obvious to me how these methods may accomplish this, unless combined with some dynamical
analysis that suggests the causality.
**We rephrased this paragraph, indicating that the tool allows one to visually inspect co-**
**occurrences of WCB outflows and jet corelines. The visualization cannot replace the**
**dynamical analysis, as mentioned above.**
Figure 13, I think the lower panel of this figure would be seriously compromised if viewed in grey
scale (you would not be able to distinguish the "coolwarm" type color palette from the grey scale
tropopause surface. You might thus want to think about changes to the presentation here.
**We changed the color map of the vertical slice to a sequential color map. In case of**
**conversion to grayscale, the shading of the tropopause can be distinguished from the**
**coloring of the vertical slice better than before.**

[Figure]

Lines 414--415, Per general comment (2b), there is already a vast body of research on these
topics, including the few papers I've mentioned here along with many others, covering topics
such as relationships of jets and tropopauses to storm tracks, extreme weather events, etc. Jet
regimes of various sorts have been defined based on characteristics of the jet stream, see
general point (2b). While I think the methods in this paper can be a very valuable addition to the
existing tools and literature aimed at more fully characterising the jets as dominant features
influencing the tropospheric circulation, it is disingenuous to state these solely as future aims.
**We removed all pointers to future work that are not directly related to the improvement of**
**our approach. That is, we now only point towards the application on longer time series,**
**improving temporal stability, and achieving order-independence (see next comment). In**
**response to Reviewer 1, we also added a brief discussion on steps towards usage in**
**operational settings.**
Lines 419--420, I'm not sure what you mean by a "non-incremental" search, perhaps you can
explain this briefly.
**The current algorithm is incremental, in the sense that it extracts one jet after the other.**
**The final result thereby becomes dependent on the order in which the jets have been**
**extracted. It would be interesting to investigate how the jet extraction could be made**
**order-independent. We rephrased this accordingly in the future work section, removing**
**the term "non-incremental".**
**Typos / Grammar / Minor Wording / Etc:**
Line 16, "is" should be "are" ("data" is plural).
**Corrected.**

Line 17, "time-dependent" should be followed by a comma.
**Corrected.**

Line 17, "This data is" -> "These data are".
**Corrected.**

Line 66, I don't think "package" is the best word here; I would just say something like "rotation of the air enclosed between two…" (in fact if there are diabatic motions, it is not "trapped" in any sense).
**Rephrased as suggested.**

Line 84, I would hardly call a paper published in 2001 "recent".
**We removed the word "recent".**

Lines 260 and 262, "is" should be "are".
**Corrected.**

Line 350, "he jet streaks" should be "the jet streaks".
**Corrected.**

Line 379, by "attained considerable focus" do you mean it is a topic currently under investigation? If so, just say that.
**We rephrased this to "is of interest to".**

Line 421, "Dur" should be "During"
**Corrected.**

**References not already cited:**
(Apologies for non-uniform format, these are pasted from most convenient sources.)

- Danielsen, E. F. (1968), Stratospheric-tropospheric exchange based on radioactivity, ozone and potential vorticity, J. Atmos. Sci., 25, 502–518, doi:10.1175/1520-0469(1968)025 <0502:STEBOR> 2.0.CO;2
- Harnik, N., C. I. Garfinkel, and O. Lachmy, 2016: The influence of jet stream regime on extreme weather events. Dynamics and Predictability of Large-Scale, High-Impact Weather and Climate Events, J. Li, Ed., Cambridge University Press, 79–94, https://doi.org/10.1017/CBO9781107775541.007.
- Highwood, E. J., B. J. Hoskins, and P. Berrisford, 2000: Properties of the Arctic tropopause. Quart. J. Roy. Meteor. Soc., 126, 1515–1532, doi:10.1002/qj.49712656515.
- Holton, J. R., P. H. Haynes, M. E. McIntyre, A. R. Douglass, R. B. Rood, and L. Pfister (1995), Stratosphere-troposphere exchange, Rev. Geophys., 33(4), 403–440, doi:10.1029/95RG02097.
- Kunz, A., P. Konopka, R. Müller, and L. L. Pan (2011), Dynamical tropopause based on isentropic potential vorticity gradients, J. Geophys. Res., 116, D01110, doi:10.1029/2010JD014343.
- Lee, S., and H.-K. Kim, 2003: The dynamical relationship between subtropical and eddy-driven jets. J. Atmos. Sci., 60, 1490–1503, doi:10.1175/1520-0469(2003)060,1490:TDRBSA.2.0.CO;2.
- Lillo, S. P., Cavallo, S. M., Parsons, D. B., & Riedel, C. (2021). The Role of a Tropopause Polar Vortex in the Generation of the January 2019 Extreme Arctic

Outbreak, Journal of the Atmospheric Sciences, 78(9), 2801-2821.
https://journals.ametsoc.org/view/journals/atsc/78/9/JAS-D-20-0285.1.xml

- Lucas, C., B. Timbal, and H. Nguyen, 2014: The expanding tropics: A critical assessment of the observational and modeling studies. Wiley Interdiscip. Rev.: Climate Change, 5, 89–112, https://doi.org/10.1002/wcc.251.
- Maher, P., M. E. Kelleher, P. G. Sansom, and J. Methven, 2020: Is the subtropical jet shifting poleward? Climate Dyn., 54, 1741–1759, https://doi.org/10.1007/s00382-019-05084-6.
- Manney, G.L., et al., Jet characterization in the upper troposphere/lower stratosphere (UTLS): Applications to climatology and transport studies, Atmos. Chem. Phys., 11, 6115–6137, 2011.
- Manney, G.L., M.I. Hegglin, W.H. Daffer, M.J. Schwartz, M.L. Santee, and S. Pawson, Climatology of Upper Tropospheric/Lower Stratospheric (UTLS) Jets and Tropopauses in MERRA, J. Clim., 27, 3248–3271, 2014.
- Manney, G.L., and M.I. Hegglin, Seasonal and Regional Variations of Long-Term Changes in Upper Tropospheric Jets from Reanalyses, J. Clim., 31, 423–448, 2018.
- Manney, G.L., Z.D. Lawrence, and M.I. Hegglin, Relationships of interannual variability in upper tropospheric jets to ENSO in reanalyses, J. Clim., https://doi.org/10.1175/JCLI-D-20-0947.1, 2021.
- Schoeberl, M. R. (2004), Extratropical stratosphere-troposphere mass exchange, J. Geophys. Res., 109, D13303, doi:10.1029/2004JD004525.
- Shapiro, M. A. (1980), Turbulent mixing within tropopause folds as a mechanism for the exchange of chemical constituents between the stratosphere and troposphere, J. Atmos. Sci., 37, 994–1004, doi:10.1175/1520-0469(1980)037 < 0994:TMWTFA > 2.0.CO;2.
- Spensberger, C., and T. Spengler, 2020: Feature-based jet variability in the upper troposphere. J. Climate, 33, 6849–6871, https://doi.org/10.1175/JCLI-D-19-0715.1.
- Stohl, A., et al. (2003), Stratosphere-troposphere exchange: A review, and what we have learned from STACCATO, J. Geophys. Res., 108(D12), 8516, doi:10.1029/2002JD002490.
- Winters, A. C., Keyser, D., Bosart, L. F., & Martin, J. E. (2020). Composite Synoptic-Scale Environments Conducive to North American Polar–Subtropical Jet Superposition Events, Monthly Weather Review, 148(5), 1987-2008. https://journals.ametsoc.org/view/journals/mwre/148/5/mwr-d-19-0353.1.xml

**We added the suggested references.**

---

## Referee Report (RR1)

**Re-evaluation of "Integration-based Extraction and Visualization of Jet Stream Cores", by Bösiger et al.** (Review by Gloria Manney)

**Recommendation:** Publish after some technical corrections

**General Comments:** This paper presents a new method and software for tracking and visualizing jet stream cores. This is a potentially very useful new method with some important advantages and should be a valuable addition to existing tools for jet characterization and analysis. The authors have done a thorough job of addressing my comments on the first version, and I believe the paper is now suitable for publication in GMD after the authors consider the following additional small changes / corrections (line numbers from ATC version):

Figures: I still think the color bars and their text labels are too small.

Line 94, suggest "...which is called the *jet core*… (add "the")

Line 109, typo, should read simply "...(Lee and Kim, 2003;..."

Lines 115–119, this is probably more detail than you need, you could simply start with "Manney et al (2011, 2014) identified…" and then leave out the sentence starting "This approach was applied…". Also you might reduce this a little further by saying something like "...inside latitude-altitude slices, with additional criteria applied to determine whether multiple maxima in the same 30 m s-1 contour represented separate jet stream cores." (If you keep the same wording for this part, note that "below" on line 117 should be "by more than".) Finally, you could delete the sentence about Manney & Hegglin's subtropical / polar jet definition here (lines 118-120) since you have already said it earlier.

Line 176, suggested wording "...other time periods, both below and above our chosen thresholds (e.g., Manney et al., 2014)."

Line 263, suggest "...or are duplicated for numerical reasons…"

Table 1 caption, first line, change "here listed" to "listed here".

Line 414, change "involved physical processes" to "physical processes involved".

Line 418, would be good to give the Maher et al (2019) and Winters et al (2020) references here along with Koch et al (2006).

Line 462, I don't think "tangible" is the right word here. I think what you mean is more like "tractable" or "managable".

Line 468–469, either say "...becomes more noticeable when too little weight is given…" or "became more noticeable when too little weight was given…"

---

## Author Response (AR2)

**Integration-based Extraction and Visualization of Jet Stream Cores**

**Corrections**

Submission ID: gmd-2021-240

**Geoscientific Model Development**

Dear Reviewers,

Again, we would like to thank you for your constructive and helpful comments. We made the suggested changes and hereby submit the revised files.

Sincerely,
The authors.

**Review by Gloria Manney**

**General Comments:**
This paper presents a new method and software for tracking and visualizing jet stream cores. This is a potentially very useful new method with some important advantages and should be a valuable addition to existing tools for jet characterization and analysis. The authors have done a thorough job of addressing my comments on the first version, and I believe the paper is now suitable for publication in GMD after the authors consider the following additional small changes / corrections (line numbers from ATC version):

Figures: I still think the color bars and their text labels are too small.
**We further increased the size of all color bars by 20% and increased the font size of their text labels from "\scriptsize" to "\small".**

Line 94, suggest "...which is called the jet core… (add "the")
**Done.**
*"The center of the jet stream, which is called the jet core [...]"*

Line 109, typo, should read simply "...(Lee and Kim, 2003;..."
**Fixed.**
*"[...] continuous spectrum of jet characteristics (Lee and Kim, 2003; [...]"*

Lines 115–119, this is probably more detail than you need, you could simply start with "Manney et al (2011, 2014) identified…" and then leave out the sentence starting "This approach was applied…". Also you might reduce this a little further by saying something like "...inside latitude-altitude slices, with additional criteria applied to determine whether multiple maxima in the same 30 m s-1 contour represented separate jet stream cores." (If you keep the same wording for this part, note that "below" on line 117 should be "by more than".) Finally, you could delete the sentence about Manney & Hegglin's subtropical / polar jet definition here (lines 118-120) since you have already said it earlier.
**We reworded the paragraph as suggested. We would like to keep the more detailed version that explains the additional criteria.**
*"Manney et al. (2011, 2014) identified [...]", "[...] drops by more than [...]", and we removed the last sentence as suggested.*

Line 176, suggested wording "...other time periods, both below and above our chosen thresholds (e.g., Manney et al., 2014)."
**Done.**
*"[...] above our chosen thresholds (e.g. Manney et al., 2014)."*

Line 263, suggest "...or are duplicated for numerical reasons…"
**Fixed.**
*"[...] or are duplicated for numerical reasons."*

Table 1 caption, first line, change "here listed" to "listed here".
**Fixed.**
*"[...], listed here for the default parameters [...]*

Line 414, change "involved physical processes" to "physical processes involved".
**Fixed.**
*"[...] dynamical analysis of the physical processes involved, [...]"*

Line 418, would be good to give the Maher et al (2019) and Winters et al (2020) references here along with Koch et al (2006).
**Added.**
*"[...] gradient (Maher et al., 2019; Winterset al., 2020; Koch et al., 2006), [...]"*

Line 462, I don't think "tangible" is the right word here. I think what you mean is more like "tractable" or "managable".
**Changed to "manageable".**
*"It becomes more manageable if [...]"*

Line 468–469, either say "...becomes more noticeable when too little weight is given…" or "became more noticeable when too little weight was given…"
**Added "more" (first suggestion)**
*"[...], which becomes more noticeable when too little weight [...]"*